# Distilling LLM Agent into Small Models with Retrieval and Code Tools

**Minki Kang**[1*]   **Jongwon Jeong**[2*]   **Seanie Lee**[1]   **Jaewoong Cho**[3]   **Sung Ju Hwang**[1,4]

[1]KAIST, [2]University of Wisconsin-Madison, [3]KRAFTON, [4]DeepAuto.ai
{minkikang, sjhwang82}@kaist.ac.kr

## Abstract

Large language models (LLMs) excel at complex reasoning tasks but remain computationally expensive, limiting their practical deployment. To address this, recent works have focused on distilling reasoning capabilities into smaller language models (sLMs) using chain-of-thought (CoT) traces from teacher LLMs. However, this approach struggles in scenarios requiring rare factual knowledge or precise computation, where sLMs often hallucinate due to limited capability. In this work, we propose **Agent Distillation**, a framework for transferring not only reasoning capability but full task-solving behavior from LLM-based agents into sLMs with retrieval and code tools. We improve agent distillation along two complementary axes: (1) we introduce a prompting method called *first-thought prefix* to enhance the quality of teacher-generated trajectories; and (2) we propose a *self-consistent action generation* for improving test-time robustness of small agents. We evaluate our method on eight reasoning tasks across factual and mathematical domains, covering both in-domain and out-of-domain generalization. Our results show that sLMs as small as 0.5B, 1.5B, 3B parameters can achieve performance competitive with next-tier larger 1.5B, 3B, 7B models fine-tuned using CoT distillation, demonstrating the potential of agent distillation for building practical, tool-using small agents. Our code is available at https://github.com/Nardien/agent-distillation.

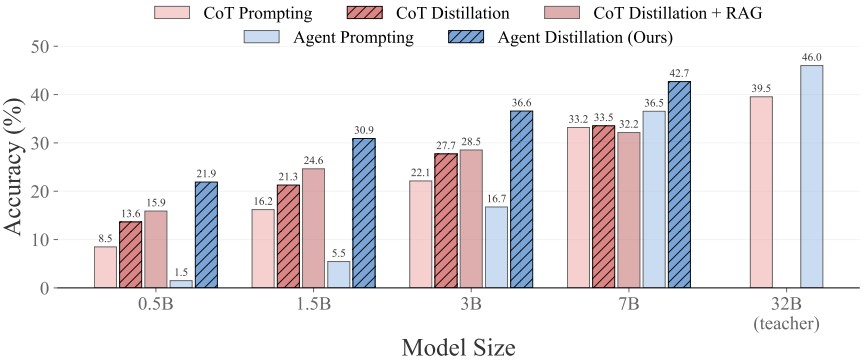

Figure 1: Performance comparison of different sizes of `Qwen2.5-Instruct` models [1] on the average accuracy of four factual reasoning tasks (HotpotQA [2], Bamboogle [3], MuSiQue [4], 2WikiMultiHopQA [5]) and four mathematical reasoning tasks (MATH [6], GSM-Hard [7], AIME [8], OlymMATH [9]). Distillation is done using the 32B model as the teacher and models ranging from 0.5B to 7B as students. Agent distillation consistently improves the performance of smaller models across both domains by enabling them to perform code execution and retrieve information for tasks adaptively. Full results are provided in Table 2.

---

*Work done at KRAFTON.

39th Conference on Neural Information Processing Systems (NeurIPS 2025).

Train query: What would $100 invested in Apple stock in 2010 be worth by 2020?

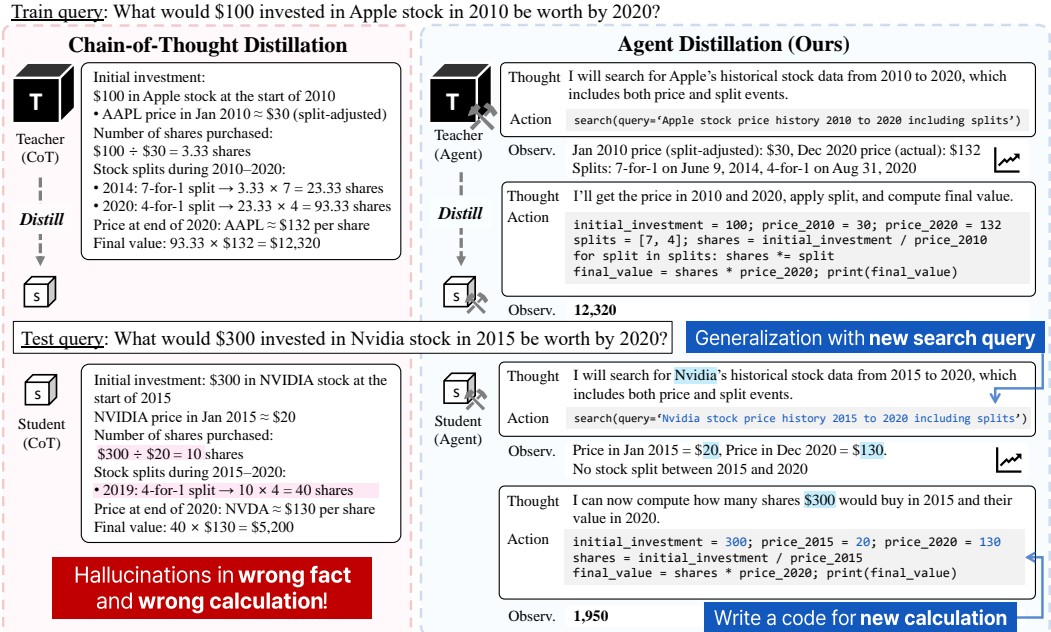

Figure 2: **Concept.** Chain-of-Thought (CoT) distillation trains student models to mimic static reasoning traces from LLMs, but often fails when new knowledge or precise computation is needed at test time. Our proposed agent distillation instead teaches student models to think and *act* (e.g., retrieve facts or execute code) offering stronger generalization and better robustness to hallucination.

# 1 Introduction

Large language models (LLMs) have achieved remarkable performance across complex real-world tasks, surpassing average human accuracy on college-level mathematics and demonstrating competence in high-stakes domains [10–12]. However, as LLM usage grows, their high inference cost becomes increasingly burdensome. While these considerations have motivated growing interest in smaller language models (sLMs) [13, 14], preserving the problem-solving capabilities of larger models in sLMs remains challenging. Therefore, a core research question emerges: *how can we preserve LLM-level problem-solving ability in much smaller models?*

Although recent advancements in pre- and post-training methods have steadily increased the capabilities of sLMs [15], sLMs still struggle to solve complex tasks at the level of LLMs. To address this, recent works have explored reasoning distillation, where sLMs are trained to mimic CoT reasoning traces generated by teacher LLMs through next-token prediction [10, 11, 1, 16, 17].

However, distilled small models are prone to hallucination and often fail to perform accurate calculations [18]. For example, answering the real-world question, *"What would $100 invested in Apple stock in 2010 be worth by 2020?"*, requires both factual knowledge about stock history and arithmetic reasoning. As illustrated in Figure 2, LLMs can correctly answer this question using CoT by leveraging memorized knowledge and numerical skills. However, simply distilling such a reasoning trace into an sLM does not guarantee generalization, especially those involving new knowledge or calculation not observed during distillation due to their limited capability [19].

In this work, we propose **Agent Distillation**, a framework that moves beyond static reasoning to distill the ability to take action with tools, from LLM agents (*e.g.,* ReAct [20], CodeAct [21]) into sLMs through *reason-act-observe* trajectories. Our goal is to equip sLMs with agentic capabilities: reasoning through problems, taking actions to use code or retrieval tools, observing outcomes, and refining their approach—cloning the behavior of LLM agents. This approach offers two key advantages: (1) sLMs focus on learning how to reason and act to solve problems using tools, rather than memorizing knowledge and calculations, and (2) they generalize better to new queries requiring previously unseen facts or calculations. A remaining challenge is whether such complex agentic behavior can be distilled from a large teacher model (>30B) into a much smaller student (0.5–3B) [1].

To this end, we introduce two simple but effective methods to aid effective distillation. First, we propose a *first-thought prefix* method that aligns agentic reasoning with the teacher model's instruction-tuned behavior, improving trajectory quality of teacher agent without additional fine-tuning. These improved trajectories offer better supervision for sLM distillation. Second, we improve student robustness at test-time through *self-consistent action generation*, which samples multiple trajectories and selects the one yielding a valid and consistent outcome leveraging code interpreter.

We evaluate our agent distillation on four factual (*e.g.,* HotPotQA [2]) and four mathematical (*e.g.,* MATH [6]) reasoning benchmarks. For each reasoning type, we consider one in-domain task and three out-of-domain tasks to test generalization. As in Figure 1, our results show that agent distillation consistently enhances the problem-solving capabilities of small models of 0.5B to 7B.

To summarize, our work makes the following key contributions:

- We propose **Agent Distillation**, a framework for training sLMs to imitate trajectories from LLM agents, enabling agentic behavior without memorizing factual knowledge and calculations.
- We introduce two methods to overcome limitations of naive distillation: (1) a *first-thought prefix* for improving teacher trajectory, and (2) *self-consistent action generation* to boost test-time robustness.
- We validate our method across 8 factual and mathematical reasoning benchmarks, showing strong performance across domains and student model scales (0.5B-7B) compared to CoT distillation.
- Remarkably, we demonstrate that **even 0.5B, 1.5B, and 3B models** distilled with our method can achieve *comparable performance to next-tier larger models* distilled with CoT on average.

## 2 Related works

### 2.1 Reasoning distillation of language models

Large language models (LLMs) have shown strong performance on complex reasoning tasks using methods like chain-of-thought (CoT) prompting [22, 23]. To transfer these capabilities to smaller models (sLMs), CoT distillation methods [16, 17, 24–27] train sLMs to reproduce step-by-step reasoning traces from stronger LLMs. This has proven effective—particularly in mathematical reasoning—and is now a common component of post-training pipelines [11, 1]. To improve generalization, recent methods incorporate external tools such as retrieval [19, 28, 29] or code execution [30–32], helping sLMs focus on transferable reasoning strategies rather than memorization of others. Still, most existing approaches rely on static demonstrations and lack interaction with the environment.

In contrast, we distill agentic behaviors where models learn the reasoning and tool use during interactions with environments. This enables sLMs to learn *how to act* for solving tasks.

### 2.2 Language agents and agentic reasoning

An agent can be broadly defined as an entity that autonomously pursues goals by observing the world and acting upon it. Powered by LLMs, early works like ReAct [20, 33] introduced the concept of *language agents*—which observe the world, *think in natural language*, and act to complete the diverse range of tasks interactively. Since most LLMs are not natively trained for such interaction, prior works have relied on carefully designed prompts (*e.g.,* few-shot examples) for stronger LLMs, and fine-tuned weaker LLMs on trajectories from stronger ones [20, 33–42]. Building on foundations, recent works have pushed language agents toward more advanced agentic capabilities.

Early works focus on teaching LLMs to use tools that enable interaction with external environments [43–47]. Furthermore, agentic retrieval systems have emerged to support multi-hop reasoning over real-world knowledge [48–50], while tool-augmented reasoning leverages external capabilities like code execution to tackle challenging math problems [51–55]. Other approaches promote the notion of *agentic reasoning*, enhancing the decision-making and planning capabilities of LLMs for solving complex tasks with tools through prompting or reinforcement learning [56–58].

**Unlike prior work**, which primarily focused on fine-tuning LLMs ($\geq$ 7B) on trajectories from stronger close-sourced LLMs (*e.g.,* GPT-4 [12] in FireAct [34]), our work aims to **distill** the agentic capabilities of LLMs into much smaller models (sLMs, $\leq$ 3B), enabling them to operate as capable agents. We address key challenges such as improving the quality of teacher trajectories and optimizing student behavior at test time, building on improved agent framework [21]. We show its effectiveness

across a range of small models (*e.g.,* 0.5B-3B) and tasks requiring strong knowledge and reasoning capabilites–an under-explored yet important setting for practical, small language agents.

## 3 Preliminary

**Knowledge Distillation.** Knowledge distillation [59] transfers the capabilities of a large teacher model $p_T$ to a smaller student model $p_S$. Modern language models follow the auto-regressive transformer architecture [60], where a token-level policy predicts the next token given previous tokens. Given source and target sequences $(\boldsymbol{x}, \boldsymbol{y})$, distillation optimizes the following objective:

$$\min_\theta \mathbb{E}_{(\boldsymbol{x},\boldsymbol{y})\sim\mathcal{D}_{\text{train}}} \frac{1}{L_{\boldsymbol{y}}} \sum_{n=1}^{L_{\boldsymbol{y}}} D(p_T(\cdot \mid \boldsymbol{y}_{<n}, \boldsymbol{x}) \parallel p_S(\cdot \mid \boldsymbol{y}_{<n}, \boldsymbol{x}; \theta)), \quad (1)$$

where $D$ is a divergence metric (*e.g.,* Kullback–Leibler or Jensen–Shannon divergence), and $L_{\boldsymbol{y}}$ denotes the length of the target sequence $\boldsymbol{y}$.

**Reasoning distillation.** In reasoning tasks, the target sequence $\boldsymbol{y}$ can be a rationale that solves the problem step-by-step. Since collecting human-annotated reasoning is expensive, recent approaches [16, 24, 25, 27] use chain-of-thought (CoT) prompting [23] to generate rationales with large teacher models and train the student to imitate them:

$$\min_\theta -\mathbb{E}_{\boldsymbol{x}\sim\mathcal{D}_{\text{train}}, \boldsymbol{y}\sim p_T(\cdot|\boldsymbol{x}, \boldsymbol{I}_{\text{CoT}})} \sum_{n=1}^{L_{\boldsymbol{y}}} \log p_S(\boldsymbol{y}_n \mid \boldsymbol{x}, \boldsymbol{y}_{<n}; \theta), \quad (2)$$

where $\boldsymbol{I}_{\text{CoT}}$ denotes a CoT-style prompt such as "Let's think step by step." [23].

## 4 Agent Distillation

While reasoning distillation is effective and has become a standard post-training technique [11, 1], it does not equip models with the ability to interact with external environments through actions. Recent work [20, 21] shows that large models can generate actions grounded in intermediate reasoning, observe feedback from the environment, and adapt accordingly.

We refer to such interactive sequences as *agent trajectories*, consisting of repeated cycles of thought ($\boldsymbol{r}$), action ($\boldsymbol{a}$), and observation ($\boldsymbol{o}$). Given an input $\boldsymbol{x}$, the teacher model generates a trajectory:

$$\tau = ((\boldsymbol{r}_1, \boldsymbol{a}_1, \boldsymbol{o}_1), \dots, (\boldsymbol{r}_{L_\tau}, \boldsymbol{a}_{L_\tau}, \boldsymbol{o}_{L_\tau})) \sim p_T(\cdot \mid \boldsymbol{x}, \boldsymbol{I}_{\text{agent}}), \quad (3)$$

where $\boldsymbol{I}_{\text{agent}}$ is an instruction prompt for the agent (*e.g.,* "To solve the task, you must plan forward to proceed in a series of steps, in a cycle of Thought:, Code:, and Observation: sequences" [21, 61]). Each observation $\boldsymbol{o}$ comes from the environment in response to action $\boldsymbol{a}$, not generated by the model.

Following prior works [34, 52], we fine-tune the student model on generated trajectories, excluding observations from the loss:

$$\min_\theta -\mathbb{E}_{\boldsymbol{x}\sim\mathcal{D}_{\text{train}}, \tau\sim\pi_T(\cdot|\boldsymbol{x}, \boldsymbol{I}_{\text{agent}})} \sum_{t=1}^{L_\tau} \log p_S(\boldsymbol{r}_t, \boldsymbol{a}_t \mid \boldsymbol{x}, \tau_{<t}; \theta), \quad (4)$$

where $\tau_{<t} = ((\boldsymbol{r}_1, \boldsymbol{a}_1, \boldsymbol{o}_1), \dots, (\boldsymbol{r}_{t-1}, \boldsymbol{a}_{t-1}, \boldsymbol{o}_{t-1}))$.

This distillation enables student models to function as interactive agents. For instance, a model distilled from CodeAct [21] can reason about which code snippet to generate, generate actions as codes (*e.g.,* API calls, loops), and respond to execution feedback. If the interpreter returns an error, the model can revise the code accordingly; if the output is valid but insufficient (*e.g.,* suboptimal search results), it can rephrase the query and continue the task adaptively.

Despite its promise, agent distillation presents two key challenges, particularly when applied to small language models (sLMs). First, agentic behavior often lies out-of-distribution relative to the pre-training and instruction-tuning distribution of both teacher and student models. As a result, distilling such behavior may degrade performance on domains where the student is already well-optimized for CoT-style reasoning. Second, although sLMs are pretrained on large code corpora [62], they may struggle to produce functional code during inference. Typical failure cases include misformatted code outputs or incorrect usage of library functions, which hinder the ability of agents to interact.

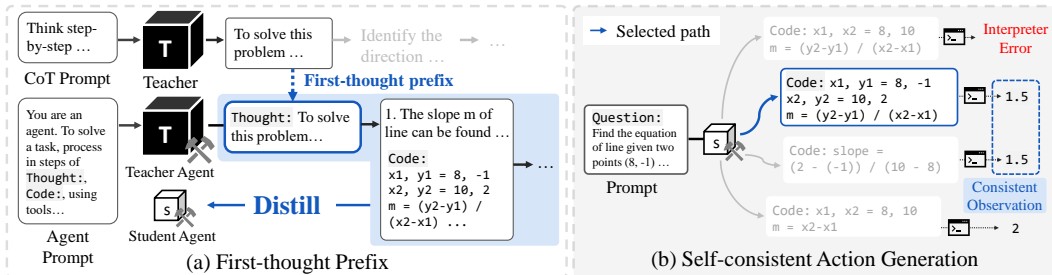

Figure 3: **(a) First-thought Prefix:** We prompt teacher with a CoT prompt to induce step-by-step reasoning. The first reasoning step is used as a prefix to generate an agentic trajectory, which is then distilled to a student agent to teach CoT-style reasoning initialization. **(b) Self-consistent Action Generation:** The agent generates multiple candidate actions and selects the one with consistent outcomes. Thoughts are omitted for brevity.

**First-thought prefix.** We observe that instruction-tuned LLMs (*e.g.*, Qwen2.5-32B-Instruct [1]), when employed as agents, demonstrate reduced performance on challenging problems from MATH500 benchmarks compared to their performance with CoT prompting [23] (see Appendix D.1 for experimental results). This degradation can further propagate during distillation, negatively impacting student models where they have also been instruction-tuned on CoT-style data.

We hypothesize that instruction-tuned models, which have already been trained to produce CoT reasoning to solve the task, can exhibit distributional drift when prompted with agent instructions (*e.g.*, Prompt D.2 in Appendix). Although these models are capable of structured reasoning, the additional instruction to generate reason-act trajectories may override or conflict with their original reasoning patterns. As a result, the model may deviate from the correct reasoning path it would otherwise follow under CoT prompting. Since prior studies have shown that the initial reasoning step critically determines the final conclusion of LLMs [63–65], ensuring that the model begins reasoning in an appropriate direction during its first action generation becomes essential for maintaining accurate reasoning.

To this end, we propose the first-thought prefix (`FTP`). Motivated by the prefix-attack in LLM jail-breaking works [66–68], this method integrates the initial reasoning step from a CoT prompting as a prefix to the agent's first thought as in Figure 3(a). Formally, we modify the trajectory sampling described in Equation 3 as follows:

$$\boldsymbol{y}_1 \sim p_T(\cdot \mid \boldsymbol{x}, \boldsymbol{I}_{\mathsf{CoT}}), \quad \tau = \{(\boldsymbol{r}_1', \boldsymbol{a}_1, \boldsymbol{o}_1), \ldots, (\boldsymbol{r}_{L_\tau}, \boldsymbol{a}_{L_\tau}, \boldsymbol{o}_{L_\tau})\} \sim p_T(\cdot \mid \boldsymbol{x}, \boldsymbol{y}_1, \boldsymbol{I}_{\mathsf{agent}}), \quad (5)$$

where $\boldsymbol{y}_1$ is the first-step of CoT reasoning and $\boldsymbol{r}_1'$ denotes the completed first thought of the agent following the prefixed first-step $\boldsymbol{y}_1$. Note that this method is **only used to generate trajectories from the teacher agent**; the student agent does not explicitly require first-thought prefix during inference.

**Self-consistent action generation.** We observe that small distilled agents often produce invalid actions, particularly in the context of CodeAct [21], where invalid actions refer to code that either fails to execute or throws errors. To improve robustness in action generation, we introduce self-consistent action generation (`SAG`). Instead of using greedy decoding, we sample multiple $N$ thought-action sequences for each step through nucleus sampling [69] with a high temperature to encourage diversity. We then filter out any sequences that result in parsing or execution errors using a lightweight code interpreter. When all generated actions fail, we retain one randomly selected failed action and feed its error message back as an observation, allowing the model to self-correct in subsequent steps. To further ensure correctness, we perform majority voting over the resulting observations [70], selecting the action whose output is most consistent across samples. For example, in Figure 3(b), the agent generates four candidate sequences. One result in an interpreter error is filtered out. Among the remaining three, two produce the same output, so we select one of these two consistent actions as a final action.

## 5  Experimental setup

We evaluate our proposed Agent Distillation across benchmarks to test whether small language models (sLMs) can acquire agentic abilities from a large language model (LLM) agent teacher.

Table 1: Task categorization with domain and sampled test data size we used.

| Task Type | Domain | Dataset Name | Description | Test Data Size |
|---|---|---|---|---|
| Factual Reasoning | In-domain | HotPotQA [2] | 2-hop question-answering | 500 |
| | Out-of-domain | Bamboogle [3] | 2-hop question-answering | 125 |
| | Out-of-domain | MuSiQue [4] | 3-hop question-answering | 500 |
| | Out-of-domain | 2WikiMultiHopQA [5] | 2-hop question-answering | 500 |
| Math Reasoning | In-domain | MATH [6] | College-level math | 500 |
| | Out-of-domain | GSM-Hard [7] | Large number arithmetics | 500 |
| | Out-of-domain | AIME [8] | Olympiad-level problems | 90 |
| | Out-of-domain | OlymMath [9] | Olympiad-level problems | 200 |

**Tasks and datasets.** We evaluate two categories of reasoning tasks: factual and mathematical. For each, we assess both in-domain and out-of-domain generalization. We use 1,000 HotPotQA [2] and 2,000 MATH [6] examples for training. For test benchmarks, we summarize them in Table 1. To reduce evaluation cost, we limit each test set to 500 examples, following Wang et al. [71]. As a metric, we use exact match for math and llm-as-a-judge [72] using `gpt-4o-mini` for factual reasoning.

**Models.** The teacher model is `Qwen2.5-32B-Instruct`, a 32B parameter instruction-tuned model. For student models, we use the `Qwen2.5-Instruct` series with four sizes: 0.5B, 1.5B, 3B, and 7B parameters. All student models are instruction-tuned prior to distillation [1].

**Baselines.** We compare two main distillation paradigms: (1) CoT distillation [16], which transfers static reasoning traces generated using Chain-of-Thought prompting, and (2) our proposed Agent Distillation, which transfers interactive reason-act-observe trajectories. For CoT distillation, we add the baseline that uses retrieval-augmented generation [73] in both distillation and inference for a fair comparison with external knowledge [19, 28, 29]. For ours, we adopt the formulation from CodeAct [21, 61], where each step consists of a `Thought`, `Action` (*e.g.,* Python code), and `Observation`. Additionally, we incorporate two proposed methods — distillation using trajectories through first-thought prefix $\overline{\texttt{FTP}}$ and self-consistent action generation $\overline{\texttt{SAG}}$.

**Training & inference details.** For reproducibility of experiments, we use Wikipedia 2018 as a knowledge base for both agents and RAG instead of search engine. We use `e5-base-v2` [74] as both document and query embeddings as in Jin et al. [49]. For both CoT and agent, we sample one trajectory per question from the teacher model and **filter out wrong trajectories**, resulting in approximately 2,000 trajectories for distillation.

We fine-tune student models using parameter-efficient tuning with LoRA (rank 64) on all linear layers [75]. All models are fine-tuned for 2 epochs using a batch size of 8 and a learning rate of $2 \cdot 10^{-4}$. All experiments are conducted using four NVIDIA A100 80GB GPUs.

For inference, we use a greedy decoding. For all agents, we set max steps to 5. For $\overline{\texttt{SAG}}$ in main experiments, we set $N = 8$ with temperature to 0.4. More details are in Appendix C.

# 6   Results

**Overall results.** In Table 2, we find that agent distillation consistently improves performance across all model sizes. Before distillation, most sizes of models (except 7B) fail to produce effective agentic outputs via prompting alone, often generating incorrect or unparseable code action. In contrast, our distilled agents outperform CoT-distilled counterparts, particularly on out-of-domain tasks across both factual and mathematical domains. These results highlight the **effectiveness of agent distillation in improving generalization of sLMs**. Notably, the gains are further amplified by our two methods–First-thought Prefix ($\overline{\texttt{FTP}}$) and Self-consistent Action Generation ($\overline{\texttt{SAG}}$).

Our findings also demonstrate that **agent distillation enables small models to match or exceed the performance of CoT-distilled models that are 2–4× larger**, offering a promising path toward efficient and capable language agents. Specifically, the 0.5B agent matches the performance of a 1.5B CoT-distilled model, the 1.5B agent reaches its 3B counterpart, the 3B agent surpasses the 7B CoT model, and the 7B agent even outperforms the 32B CoT model.

Table 2: **Main results.** Distilled agents show the strong performance on most of tasks, especially on out-of-domain tasks, compared to baselines. FTP = First-Thought Prefix, SAG = Self-consistent Action Generation. Highlighting best among same-sized models. Avg. denotes the average score across all tasks.

| Params | Method | | In-domain | | Out-of-domain | | | | | | |
|---|---|---|---|---|---|---|---|---|---|---|---|
| | | | HotPot QA | MATH 500 | MuSi-Que | Bamb-oogle | 2Wiki QA | GSM-Hard | AIME | Olym-MATH | Avg. |
| | | | Teacher: Qwen-2.5-Instruct | | | | | | | | |
| 32B | CoT | Prompting | 36.8 | 79.2 | 12.2 | 60.8 | 33.4 | 74.6 | 13.3 | 6.0 | 39.54 |
| | Agent | Prompting | 56.4 | 69.2 | 25.2 | 58.4 | 49.8 | 76.4 | 21.1 | 11.5 | 46.00 |
| | | | Student: Qwen-2.5-Instruct | | | | | | | | |
| 7B | CoT | Prompting | 29.2 | 71.8 | 5.8 | 43.2 | 29.2 | 66.6 | 12.2 | 7.5 | 33.19 |
| | | Distill | 31.0 | 72.6 | 9.0 | 44.8 | 26.8 | 67.6 | 10.0 | 6.5 | 33.54 |
| | | Distill + RAG | 42.8 | 68.0 | 6.6 | 40.0 | 27.6 | 60.6 | 6.7 | 5.0 | 32.16 |
| | Agent | Prompting | 46.8 | 56.0 | 16.8 | 41.6 | 45.6 | 62.2 | 13.3 | 10.0 | 36.54 |
| | | Distill | 51.2 | 62.2 | 19.6 | 52.0 | 45.2 | 72.0 | 11.1 | 5.5 | 39.85 |
| | | + FTP | 55.0 | 66.6 | 17.6 | 56.0 | 44.6 | 70.8 | 14.4 | 13.0 | 42.26 |
| | | + SAG | 53.2 | 64.0 | 20.6 | 50.4 | 48.2 | 73.4 | 15.6 | 9.5 | 41.86 |
| | | + FTPSAG | 54.4 | 67.8 | 19.4 | 55.2 | 45.2 | 72.4 | 15.6 | 11.5 | **42.68** |
| 3B | CoT | Prompting | 38.6 | 62.8 | 6.2 | 33.6 | 21.6 | 60.2 | 6.7 | 4.5 | 29.27 |
| | | Distill | 26.8 | 61.8 | 6.4 | 34.4 | 25.0 | 56.8 | 5.6 | 5.0 | 27.72 |
| | | Distill + RAG | 40.6 | 59.6 | 4.6 | 32.0 | 28.2 | 53.2 | 5.6 | 4.5 | 28.53 |
| | Agent | Prompting | 38.6 | 30.5 | 8.8 | 29.6 | 28.8 | 25.8 | 4.4 | 3.0 | 21.20 |
| | | Distill (Ours) | 48.4 | 54.0 | 13.0 | 37.6 | 37.4 | 64.2 | 6.7 | 7.5 | 33.60 |
| | | + FTP | 47.6 | 54.4 | 13.0 | 43.2 | 41.4 | 63.0 | 7.8 | 5.5 | 34.49 |
| | | + SAG | 48.6 | 57.4 | 13.0 | 36.0 | 37.4 | 65.6 | 0.0 | 10.0 | 33.50 |
| | | + FTPSAG | 49.4 | 60.2 | 15.8 | 38.4 | 41.0 | 65.4 | 15.6 | 7.0 | **36.60** |
| 1.5B | CoT | Prompting | 17.8 | 47.6 | 3.0 | 21.6 | 19.0 | 49.0 | 1.1 | 3.5 | 20.33 |
| | | Distill | 23.8 | 46.4 | 2.0 | 21.6 | 18.4 | 51.0 | 5.6 | 1.5 | 21.28 |
| | | Distill + RAG | 37.6 | 48.6 | 4.2 | 26.4 | 27.0 | 48.6 | 2.2 | 2.5 | 24.64 |
| | Agent | Prompting | 8.6 | 22.2 | 1.6 | 10.4 | 10.6 | 9.0 | 1.1 | 0.0 | 7.94 |
| | | Distill (Ours) | 43.0 | 46.8 | 9.0 | 27.2 | 35.6 | 54.8 | 1.1 | 7.0 | 28.06 |
| | | + FTP | 43.6 | 46.4 | 8.0 | 30.4 | 32.6 | 60.6 | 7.8 | 3.5 | 29.11 |
| | | + SAG | 43.8 | 49.8 | 11.6 | 31.2 | 36.6 | 58.0 | 7.8 | 3.5 | 30.29 |
| | | + FTPSAG | 45.6 | 50.6 | 9.2 | 33.6 | 33.6 | 60.6 | 6.7 | 4.5 | **30.55** |
| 0.5B | CoT | Prompting | 9.2 | 28.4 | 0.2 | 7.2 | 12.8 | 25.6 | 1.1 | 4.0 | 11.06 |
| | | Distill | 13.2 | 28.6 | 1.4 | 10.4 | 23.8 | 28.6 | 1.1 | 2.0 | 13.64 |
| | | Distill + RAG | 29.2 | 28.0 | 1.6 | 13.6 | 25.4 | 27.4 | 0.0 | 2.0 | 15.90 |
| | Agent | Prompting | 2.4 | 3.0 | 0.0 | 0.8 | 2.8 | 5.4 | 0.0 | 0.0 | 1.80 |
| | | Distill (Ours) | 34.6 | 30.4 | 7.0 | 17.6 | 28.8 | 31.2 | 3.3 | 1.0 | 19.24 |
| | | + FTP | 32.4 | 28.8 | 3.4 | 24.0 | 30.8 | 36.4 | 1.1 | 3.0 | 19.99 |
| | | + SAG | 34.0 | 33.8 | 8.2 | 13.6 | 33.0 | 33.0 | 4.4 | 0.0 | 20.01 |
| | | + FTPSAG | 33.4 | 34.4 | 5.6 | 24.0 | 31.2 | 40.8 | 3.3 | 2.5 | **21.90** |

Table 3: Comparison of performance across general and code-specific models. 32B/1.5B denote general models and 32B-Coder/1.5B-Coder denote code-specific models. For all models, we apply SAG with $N = 8$.

| Teacher | Student | HotPot QA | MATH 500 | MuSi-Que | Bamb-oogle | 2Wiki QA | GSM-Hard | AIME | Olym-MATH | Avg. |
|---|---|---|---|---|---|---|---|---|---|---|
| 32B | 1.5B | 45.6 | 50.6 | 9.2 | 33.6 | 33.6 | 60.6 | 6.7 | 4.5 | 30.55 |
| 32B-Coder | 1.5B | 42.6 | 51.4 | 10.0 | 36.8 | 36.8 | 60.0 | 6.7 | 3.0 | **30.91** |
| 32B | 1.5B-Coder | 37.8 | 52.6 | 8.2 | 30.4 | 38.0 | 59.8 | 3.3 | 6.0 | 29.52 |
| 32B-Coder | 1.5B-Coder | 41.4 | 49.2 | 9.4 | 30.4 | 37.4 | 63.6 | 4.4 | 5.5 | 30.17 |

**Factual reasoning results.** We find that retrieval improves the performance of CoT-distilled models on factual reasoning benchmarks. However, due to its static nature, it can degrade performance on tasks requiring dynamic or adaptive information use, such as mathematical reasoning. In contrast, our distilled agents outperform even RAG-enhanced CoT models. This is because agent distillation equips the model to actively retrieve and integrate knowledge during reasoning, rather than relying solely on pre-fetched documents that may be insufficient or misaligned with the task.

**Math reasoning results.** On mathematical reasoning tasks, our distilled agents demonstrate strong overall performance. The 1.5B, 3B, and 7B models show improvements on the AIME and Olym-

Table 4: Comparison of performance across `Llama-3.2-1B-Instruct` [11] and `Phi-4-mini-instruct` [76] models. Teacher model is `Qwen-2.5-32B-Instruct`. Performance trends are consistent to results in Table 2.

| Student Model | Method | | HotPot QA | MATH 500 | MuSi-Que | Bamb-oogle | 2Wiki QA | GSM-Hard | AIME | Olym-MATH | Avg. |
|---|---|---|---|---|---|---|---|---|---|---|---|
| **Llama-3.2-1B -Instruct** | CoT | Prompting | 13.2 | 28.8 | 1.2 | 14.4 | 8.0 | 19.0 | 1.1 | 2.5 | 11.53 |
| | | Distill | 18.2 | 25.6 | 2.6 | 25.6 | 19.0 | 13.8 | 1.1 | 2.0 | 13.23 |
| | Agent | FT | 36.0 | 34.6 | 2.6 | 11.2 | 26.4 | 40.4 | 1.1 | 2.0 | 19.54 |
| | | + FTP | 37.6 | 32.8 | 3.6 | 24.0 | 30.8 | 45.0 | 1.1 | 1.5 | 22.93 |
| | | + FTPSAG | 40.6 | 40.0 | 3.2 | 23.2 | 30.0 | 47.8 | 1.1 | 3.0 | **23.97** |
| **Phi-4-mini -instruct** (3.8B) | CoT | Prompting | 24.2 | 53.8 | 6.0 | 38.4 | 24.2 | 49.6 | 5.6 | 4.5 | 25.04 |
| | | Distill | 24.4 | 63.2 | 5.8 | 33.6 | 24.8 | 54.8 | 6.7 | 7.0 | 27.41 |
| | Agent | Distill | 48.2 | 52.4 | 8.8 | 27.2 | 33.6 | 69.4 | 5.6 | 6.0 | 31.52 |
| | | + FTP | 45.2 | 60.0 | 7.2 | 34.4 | 39.2 | 71.2 | 10.0 | 7.5 | 34.58 |
| | | + FTPSAG | 47.0 | 65.6 | 9.6 | 32.0 | 41.0 | 73.0 | 11.1 | 7.0 | **35.79** |

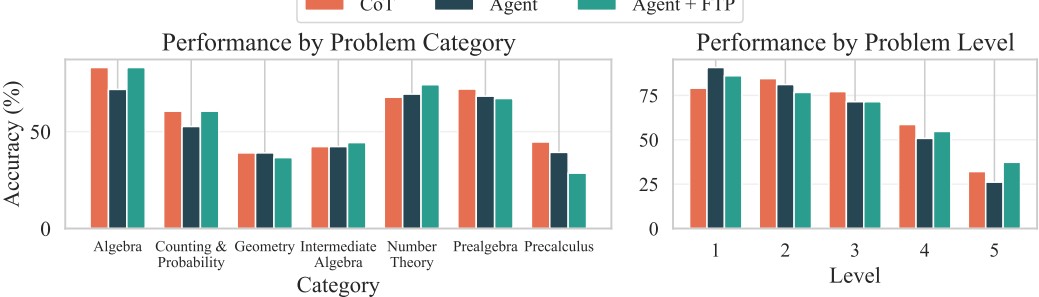

Figure 4: **Performance comparison on the MATH subcategories and levels** between CoT and Agent distillation of 3B models. Left: Accuracy by problem category. Right: Accuracy by problem difficulty level. The results highlight that FTP improves the performance of small agents in harder problems.

MATH benchmarks, benefiting from code tool use for complex calculations acquired through distillation. On GSM-hard, agent distillation improves robustness in reasoning over rare number combinations, such as 6-digits arithmetic. While performance on MATH500 lags behind CoT-distilled models in 3B and 7B models, we attribute this to the Qwen2.5 series being heavily instruction-tuned on college-level math, which may align better with CoT. Furthermore, we conjecture that larger models (3B and 7B) possess stronger internal computation skills, making tool use less beneficial on benchmarks like MATH500, while smaller models benefit more from external code execution. Nonetheless, agent distillation remains effective for larger models on harder math problems (e.g., GSM-Hard, OlymMATH), where the agentic method yields consistent gains. Overall, agent distillation delivers substantial gains across a wide range of math tasks. We provide a detailed breakdown in Section 7.

# 7 Analysis

**Code-specific teacher yields better students—marginally.** We primarily study general instruction-tuned models for both the teacher and student agents, as shown in Table 2. Given that CodeAct [21] requires generating code to perform actions, a natural question arises: Can we obtain better agents by using code-specific models for the teacher or student in the agent distillation process?

To explore this, we conduct the same set of experiments using `Qwen2.5-Coder-32B-Instruct` as the teacher and `Qwen2.5-Coder-1.5B-Instruct` as the student [77]. The results, presented in Table 3, suggest that the use of a code-specific student model does not significantly impact performance. Instead, the choice of a code-specific model as the teacher appears to be more influential in generating effective trajectories for distillation. Nevertheless, the overall improvements are marginal on average, indicating that code-specific post-training has limited impact, which suggests the code knowledge is not critical bottleneck of the student.

**Agent distillation applies across different model families.** We further validate whether the improvements from our method generalize across different language model families. In Table 4, we conduct experiments with two additional student models, `Llama-3.2-1B-Instruct` [11] and `Phi-4-mini-instruct` [76]. Results show that both models benefit from agent distillation com-

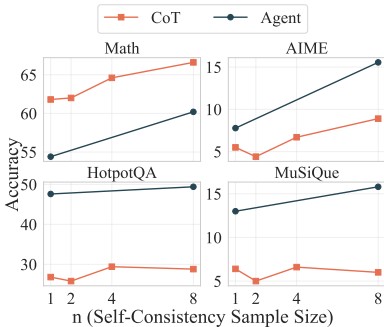

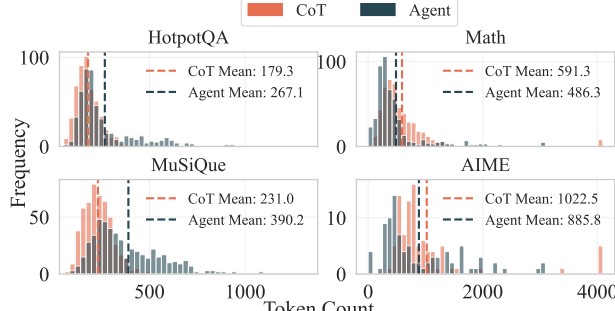

Figure 5: Comparison of `SAG` in agents and **self-consistency** [70] **in CoT** for 3B models: self-consistency in CoT is helpful in math tasks but not in factual tasks.

Figure 6: **Generated token counts** comparisons in 3B models. For factual reasoning tasks (HotpotQA, MuSiQue), agent generates more tokens than CoT. In contrast, for math reasoning tasks (MATH, AIME), CoT generates slightly more tokens than agent.

pared to CoT distillation. Moreover, both `FTP` and `SAG` yield consistent improvements across the two models, demonstrating that our proposed method is broadly applicable to different model families.

**First-thought prefix improves the agents on more complex reasoning problems.** In Table 2, we observe that agent distillation does not improve performance on MATH500 compared to CoT distillation, particularly for the 3B model. To investigate further, we break down MATH500 performance by both problem category and difficulty level.

Interestingly, naive distillation degrades the performance of distilled 3B agent on most of levels. However, when using teacher trajectories with a first-thought prefix, distilled 3B agent shows improved performance on level 4 and 5 problems–**with especially significant gains at level 5.** These results suggest that trajectories from `FTP` help student agents become more robust on complex reasoning tasks, a trend also observed in the challenging AIME benchmark in Table 2.

However, a remaining concern is the performance drop in certain categories—most notably, a decline in precalculus. Our analysis suggests that this degradation is primarily due to the nature of certain problem types that require an analytic approach rather than straightforward calculations (*e.g.,* applying properties of trigonometric functions). Such problems are harder to solve using code tools. We explore this issue in detail in Appendix D.

**Self-consistency improves CoT, but the agent with SAG still performs better.** Self-consistent action generation (`SAG`) enhances small agents by filtering out invalid code actions and retaining only those that are consistent with observations. Similarly, self-consistency [70] can be applied at test time in Chain-of-Thought (CoT) reasoning to improve performance without relying on an external verifier.

A natural question is whether CoT with self-consistency, using the same computational budget, can outperform an agent with `SAG`. To investigate this, we conduct experiments using self-consistency [70] on CoT-distilled small language models (sLMs), applying majority voting over multiple samples.

As shown in Figure 5, in the MATH benchmark—where CoT already surpasses the agent with `SAG`—self-consistency further improves the performance of the CoT-distilled model. However, in the more challenging AIME benchmark, the small agent with `SAG` still outperforms the CoT-distilled model under the same generation budget. Moreover, in factual reasoning tasks such as HotpotQA and MuSiQue, self-consistency yields only marginal gains, suggesting limited effectiveness in these settings.

**How many tokens should agents generate?** A natural question is whether a distilled agent should generate significantly more tokens than a CoT-distilled model, potentially affecting the efficiency and practicality of small models. To investigate this, we analyze token counts on two factual and two math reasoning tasks using 3B distilled models.

As shown in Figure 6, there is no significant difference in total token generation between the two approaches across both domains. In factual reasoning, the agent tends to generate more tokens due to making multiple retrieval calls across several steps to gather accurate information. In contrast, in math reasoning, the agent generates fewer tokens than CoT models by delegating repetitive calculations to code execution, often leveraging logical structures like for-loops.

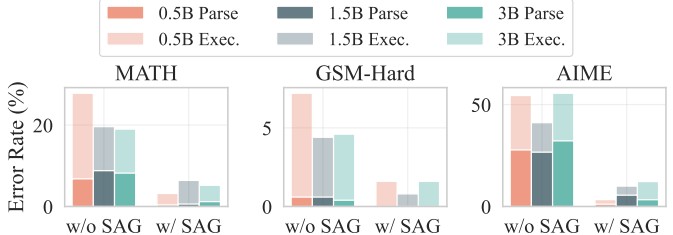
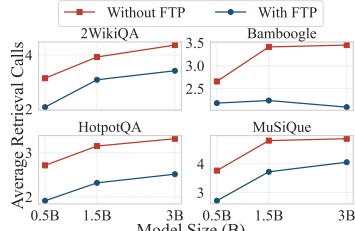

Figure 7: **Impact of Self-consistent Action Generation** (`SAG`) on code generation errors across models and 3 math datasets. `SAG` consistently reduces code parse (dark) and code execution (light) errors, especially for smaller models (0.5B) and the AIME dataset.

Figure 8: **Average retrieval tool calls** across three model sizes and datasets. Harder tasks and larger sizes make agents use more retrieval calls.

**SAG significantly reduces invalid code actions.** In Figure 7, we show the effect of self-consistent action generation (`SAG`). `SAG` reduces the generation of codes with both parsing and code execution errors. This result indicates that the small distilled agent is capable of generating valid code but the likelihood of generating valid code tends to decrease with smaller model sizes. `SAG` mitigates this issue by sampling multiple actions per turn, increasing the likelihood of generating a valid one. Nevertheless, execution errors may still occur, in which case the agent uses the error message as feedback to revise its code in the next turn.

**Larger agents make more retrieval calls, FTP reduces them** We analyze how frequently agents use the retrieval tool across different model sizes and factual reasoning benchmarks. As shown in Figure 8, larger models tend to make more retrieval calls than smaller ones, likely because they are better distilled from teacher trajectories and more effective at formulating queries and deciding when to retrieve information. In contrast, smaller models may underuse retrieval due to weaker judgment or limited capacity. For instance, they often over-rely on an initially retrieved document, even when it lacks the necessary information, rather than attempting a new retrieval.

Interestingly, we find that the first-token prefix (`FTP`) leads agents to make fewer retrieval calls. As shown in Table 2, `FTP` improves performance in Bamboogle, but results are mixed in HotpotQA and MuSiQue due to reduced retrieval. One explanation is that `FTP` encourages generating factual statements in thought process, which can lead agents—especially smaller ones—to utilize their internal knowledge instead of retrieving them, increasing the risk of hallucination. These findings suggest that the composition of teacher trajectories plays a crucial role in helping student models learn effective tool use, especially for solving complex tasks. We include more analysis in Appendix D.

# 8 Conclusion

We proposed **Agent Distillation**, a framework for transferring agentic behavior and tool use from LLMs to small language models (sLMs). By introducing first-thought prefix and self-consistent action generation, we improve both the quality of teacher trajectories and student robustness at test time. Our experiments show that distilled small agents can match or outperform next-tier larger models trained via CoT distillation, especially on out-of-domain tasks. These results highlight agent distillation as a practical path for building capable, tool-using small models for real-world problems.

**Limitations & Future Works.** While our method shows strong overall performance, it also highlights several open challenges. The first-thought prefix (`FTP`) improves agent distillation on average, underscoring the importance of high-quality teacher trajectory generation for effective distillation. However, `FTP` can sometimes degrade performance, especially when the model generates facts during reasoning instead of leveraging tools (Figure 8). This highlights the need for improved agentic trajectory generation strategies that align with the behavior and limitations of small models.

The success of self-consistent action generation (`SAG`) (Figure 7) suggests the potential of test-time compute scaling and opens up opportunities for incorporating process-level reward models [78, 79].

Finally, while agent distillation enhances the sLMs through agentic behavior, it does not directly improve their core reasoning abilities. Reinforcement learning in tool-augmented environments [48, 54, 80] could further refine these models post-distillation across diverse domains.

## Acknowledgment

This work was supported by the Institute of Information & Communications Technology Planning & Evaluation (IITP) grant funded by the Korea government(MSIT) (RS-2019-II190075, Artificial Intelligence Graduate School Program (KAIST)), the Institute of Information & Communications Technology Planning & Evaluation (IITP) with a grant funded by the Ministry of Science and ICT (MSIT) of the Republic of Korea in connection with the Global AI Frontier Lab International Collaborative Research. (No. RS-2024-00469482 & RS-2024-00509279), National Research Foundation of Korea (NRF) grant funded by the Korea government (MSIT) (No. RS-2023-00256259), and the KRAFTON AI Research Center.

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

# A  Limitations

In addition to the discussion in Section 8, we outline here several additional limitations of our study.

First, our experiments are limited to the Qwen2.5 model series [1]. While we expect our proposed approach to generalize across model families, we have not validated its effectiveness on other widely-used language models such as LLaMA [11] or Gemma [81]. Extending our study to these models would strengthen the generality of our findings and remains an important direction for future work.

Second, we only distill from a single teacher model (Qwen2.5-32B). Using stronger or larger teacher models—particularly proprietary closed-source models like GPT-4 [12]—may lead to further performance gains in student agents. However, such experiments were not feasible due to computational and budget constraints.

Third, we do not investigate the effect of the number of teacher trajectories per question on student performance, which has been shown to be an important factor in prior CoT distillation research [16, 19]. Exploring this variable could offer further insights into how to optimize agent distillation.

Lastly, our current work focuses exclusively on agents that utilize retrieval and code execution tools to solve real-world problems that the general LLM can solve without tools. Other agent applications–such as embodied agents [82] or web-based agents [83]—remain unexplored. Future research could extend agent distillation to these broader settings, leveraging tool-augmented environments such as web browsers, simulators, or desktop interfaces. In particular, integration with frameworks like the Model Context Protocol (MCP) [84] could further enhance the capabilities of small agents across diverse real-world tasks. Furthermore, ensuring safety during code execution is crucial, as unsafe operations generated by small language model agents can be irreversible or harmful. A promising direction is to apply safety-tuned decoding to reduce the likelihood of generating unsafe code or to execute code within sandboxed environments such as Docker or E2B [85].

# B  Broader impacts

This work contributes toward the development of small language agents capable of running on local devices, enabling functional on-device AI that can retrieve information from external knowledge sources (including the web) and perform code-based action to complete complex tasks.

On the positive side, this advancement promotes more accessible and inclusive AI by lowering the hardware and computational barriers for deployment. It opens opportunities for broader adoption of AI agents in resource-constrained or privacy-sensitive domains, such as healthcare, where data locality and privacy are critical.

However, there are potential risks. Since our distilled agents are capable of retrieving web information and executing code, they could be misused to automate malicious behaviors, such as generating harmful scripts or launching unauthorized attacks. Addressing these concerns will require the integration of robust safeguards, including behavior monitoring, tool-use restrictions, and secure deployment practices. We highlight this as an important avenue for future research and responsible development.

# C  Implementation details

**Prompts and agent framework.**   For CoT prompt, we use the prompt in Prompt D.1 for both math and factual reasoning. For agent prompt, we use the prompt from `smolagents` library [61]. We present the part of prompt in Prompt D.2.

As an agent framework, we use the CodeAct [21] implemented in `smolagents`. We only include the retriever for wikipedia as a tool with the name of `web_search`.

For student model, we use the same prompt for CoT reasoning. For agent, we only remove few-shot demonstrations as it is no longer needed after fine-tuning.

**Training dataset details.**   We use 1000 HotPotQA [2] and 2000 MATH [6] examples for training. Specifically, we only use 1000 hard examples from HotPotQA and 1000 level 2-3 examples, 1000 level 4-5 examples from MATH dataset. We prompt LLM to generate trajectories for both CoT

Table 5: Comparison of CoT and Agent approaches on Qwen2.5-32B-Instruct across training dataset. `FTP` denotes the first-thought prefix. Hard denotes level 4-5 and medium denotes level 2-3 questions.

| | **Model** | HotpotQA | MATH (hard) | MATH (medium) |
|---|---|---|---|---|
| CoT | Qwen2.5-32B-Instruct | 40.9 | **71.1** | **89.8** |
| Agent | Qwen2.5-32B-Instruct | 59.3 | 58.4 | 78.4 |
| | Qwen2.5-32B-Instruct + `FTP` | **60.8** | 67.1 | 83.4 |

and agent and filter out wrong trajectories based on the correctness of predicted answer. After filtering, we use approximately 2,000 trajectories to train the small models. The exact number varies depending on the performance of the teacher models on the training dataset, which we present details in Appendix D.1.

# D  Additional analysis

## D.1  Teacher model performance on training dataset

In Section 4, we propose that the first-thought prefix improves teacher performance on hard math problems. To support this, we present teacher model results on the training set in Table 5. We observe that the LLM agent outperforms a chain-of-thought (CoT) prompted LLM in factual reasoning, as the LLM relies heavily on prompting to use tools effectively—and proper tool use contributes significantly to performance. However, the performance of the LLM agent on math tasks drops considerably, especially on harder (level 4–5) problems.

In such cases, adding the first-thought prefix helps recover some of the lost performance, as discussed in Section 4. These results suggest that simply prepending the first CoT step to the agent's reasoning improves its capabilities, which in turn benefits distillation, as shown in Table 2.

## D.2  Failure case analysis of agent on the math subcategory

In example D.1, we present a failure case of the distilled 3B agent on a level 2 precalculus problem. In this instance, the generated code produces a decimal result, which is not the correct form for an answer expected in radians. Although the agent attempts a conversion in its reasoning, it ultimately produces an incorrect radian value.

Examples D.2 and D.3 involve more challenging level 4 precalculus problems. In Example D.2, for instance, the agent makes a conceptual error in its reasoning by misidentifying the appropriate range for the angle $\theta$.

These examples suggest that the agent struggles particularly with problems requiring analytic reasoning—such as understanding the properties of trigonometric functions—rather than straightforward computation.

## D.3  Deeper analysis on the first-thought prefix

**Effects on mathematical reasoning.**   As discussed in Section 4, the inclusion of a first-thought prefix (`FTP`) influences the initial reasoning patterns of the agent. In this section, we analyze how this prefix affects student agents distilled from trajectories both with and without the `FTP`, using representative examples.

In examples D.4 and D.5, drawn from the MATH500 dataset, we compare the reasoning approaches of distilled 3B agents with and without the `FTP`. Without the prefix (Example D.4), the agent's initial reasoning begins with a descriptive analysis, *e.g.,* "The problem is asking...," focusing on understanding the question. In contrast, with the prefix (Example D.5), the agent begins with a goal-oriented plan, *e.g.,* "To find the smallest positive real number...," which mirrors a chain-of-thought (CoT) strategy.

This shift illustrates that the `FTP` nudges the agent toward a more proactive and structured reasoning style, which might be beneficial in domains requiring multi-step reasoning (e.g., challenging math problems).

Table 6: Effect of temperature on math reasoning performance after agent distillation. The experiments are done with `Qwen2.5-1.5B-Instruct` with both `FTP` and `SAG`. Bold numbers indicate the best results in each column.

| Temperature | MATH500 | GSM-Hard | AIME | OlymMATH | Avg (Math) |
|---|---|---|---|---|---|
| 0.2 | 48.0 | 60.2 | **7.8** | 3.5 | 29.87 |
| 0.4 | 50.6 | 60.6 | 6.7 | **4.5** | 30.59 |
| 0.6 | 50.8 | 61.8 | 4.4 | **4.5** | 30.39 |
| 0.8 | **52.4** | 61.8 | 4.4 | 3.5 | 30.54 |
| 1.0 | 51.0 | **63.8** | 6.7 | 3.5 | **31.24** |

Table 7: Average and standard deviation across 5 different seeds in inference for agent distilled Qwen2.5-Instruct model scales on AIME with both `FTP` and `SAG`.

| Model | Avg | Std |
|---|---|---|
| 0.5B | 2.00 | 0.93 |
| 1.5B | 6.23 | 0.61 |
| 3B | 14.44 | 1.36 |

**Effects in factual reasoning.** As shown in Figure 8, the use of the first-thought prefix (`FTP`) consistently reduces the number of retrieval tool calls made by distilled agents. To better understand this phenomenon, we include illustrative examples from the Bamboogle dataset.

Examples D.6 and D.7 demonstrate cases where the `FTP` causes the distilled agent to generate factual knowledge internally rather than retrieving it. This question requires identifying the founder of geometry, the city associated with that individual, and the founder of that city.

In Example D.6, the agent (with `FTP`) directly generates the statement "The founder of geometry, Euclid" without making a retrieval call. In contrast, in Example D.7, the agent (without `FTP`) uses the retrieval tool to search for the founder of geometry, which reduces the risk of hallucination.

This pattern helps explain the behavior observed in Figure 8: while `FTP` can reduce the number of tool calls, it may also increase the likelihood of factual errors due to hallucination, as the agent relies more on internally generated knowledge.

### D.4 Deeper analysis on the self-consistent action generation

**Temperature ablation** To examine the effect of sampling temperature in self-consistent action generation (`SAG`), we evaluate the distilled `Qwen2.5-1.5B-Instruct` student model across temperatures of 0.2, 0.4, 0.6, 0.8, and 1.0 on MATH500, GSM-Hard, AIME, and OlymMATH. As shown in Table 6, performance remains relatively stable across all settings, with variations within roughly 2%. While higher temperatures (e.g., $T = 1.0$) slightly improve average accuracy by increasing action diversity, lower values (e.g., $T = 0.4$) also yield comparably strong results. These findings suggest that `SAG` is insensitive to the precise temperature choice, and we adopt $T = 0.4$ in the main experiments as a balanced configuration between diversity and reliability.

**Variance analysis** Since we stochastically sample trajectories from the model in `SAG`, randomness can introduce variation in evaluation results. This effect can be particularly noticeable for AIME, which contains only 90 questions and thus can exhibit higher variance due to its small size. To verify that our method yields consistent performance regardless of randomness (*e.g.,* random seed), we conduct inference five times with different random seeds on AIME. As shown in Table 7, the observed variance is small, corresponding to only one or two questions of difference across runs.

### D.5 Full fine-tuning vs. LoRA

All of our main experiments employ LoRA [75], owing to its low memory footprint and ease of deployment through compact adapter weights. To assess whether full fine-tuning can offer additional benefits in terms of performance, we fine-tune the `Qwen2.5-1.5B-Instruct` model for two epochs with a learning rate of $1 \times 10^{-5}$ fixing other hyperparameters unchanged compared to LoRA. As shown in Table 8, full fine-tuning yields lower average performance than LoRA-based training. While

Table 8: Comparison between LoRA and full fine-tuning (FT) on `Qwen2.5-1.5B-Instruct`.

| Method | Hotpot. | MATH | MuSiQue | Bamb. | 2Wiki. | GSM-H. | AIME | Olym. | Avg. |
|--------|---------|------|---------|-------|--------|--------|------|-------|------|
| Agent Distill (LoRA) | **43.6** | **46.4** | **8.0** | **30.4** | 32.6 | **60.6** | **7.78** | 3.5 | **29.11** |
| Agent Distill (Full FT) | 40.6 | 45.2 | 6.2 | 20.0 | **35.0** | 52.0 | 4.44 | **6.5** | 26.24 |

further hyperparameter tuning may improve the results, this trend suggests that full fine-tuning is more prone to overfitting and generalizes less effectively, making parameter-efficient adaptation preferable for agent distillation.

## Prompt D.1: Prompt for Chain-of-Thought Reasoning

You are an expert assistant who can answer the given question accurately and provide clear reasoning.

When answering questions, follow these guidelines:

1. Provide a clear and structured reasoning first
2. Follow up with a final answer, must in the <answer> </answer> tag. For example, <answer> xxx </answer>.
3. The answer must be succinct and final. For math problems, return the answer using LaTeX in the \boxed format.
4. If the question requires multiple steps or facts, break down your reasoning accordingly
5. Be precise and factual in your responses
6. If you're unsure about something, acknowledge the uncertainty

Now, please answer the following question:

## Prompt D.2: Prompt for Agent

You are an expert assistant who can solve any task using code blobs. You will be given a task to solve as best you can.
To do so, you have been given access to a list of tools: these tools are basically Python functions which you can call with code.
To solve the task, you must plan forward to proceed in a series of steps, in a cycle of `Thought:`, `Code:`, and `Observation:` sequences.

At each step, in the `Thought:` sequence, you should first explain your reasoning towards solving the task and the tools that you want to use.
Then in the `Code:` sequence, you should write the code in simple Python. The code sequence must end with `<end_code>` sequence.
During each intermediate step, you can use `print()` to save whatever important information you will then need.
These print outputs will then appear in the `Observation:` field, which will be available as input for the next step.
In the end you have to return a final answer using the `final_answer` tool.
For math problems, if not specified, always return LaTex format as the final answer.
... (ellipsis) ...
Here are the rules you should always follow to solve your task:
1. Always provide a `Thought:` sequence, and a `Code:```py` sequence ending with ```` ```<end_code> ```` sequence, else you will fail.
2. Use only variables that you have defined!
3. Always use the right arguments for the tools. DO NOT pass the arguments as a dict as in `answer = wiki('query': "What is the place where James Bond lives?")`, but use the arguments directly as in `answer = wiki(query="What is the place where James Bond lives?")`.
4. Take care to not chain too many sequential tool calls in the same code block, especially when the output format is unpredictable. For instance, a call to search has an unpredictable return format, so do not have another tool call that depends on its output in the same block: rather output results with print() to use them in the next block.
5. Call a tool only when needed, and never re-do a tool call that you previously did with the exact same parameters.
6. Don't name any new variable with the same name as a tool: for instance don't name a variable `final_answer`.
7. Never create any notional variables in our code, as having these in your logs will derail you from the true variables.
... (ellipsis) ...

**Example D.1: Failure case 1 of Agent in Precalculus (level 2)**

**Problem:** Compute $\arcsin\left(-\frac{1}{2}\right)$. Express your answer in radians.

---

**Agent:**
Thought: To compute $\arcsin\left(-\frac{1}{2}\right)$, we need to find the angle $\theta$ such that $\sin(\theta) = -\frac{1}{2}$ and $\theta$ lies within the range of the arcsine function, which is $\left[-\frac{\pi}{2}, \frac{\pi}{2}\right]$.
Code:

```python
import math

# Calculate the arcsine of -1/2
theta = math.asin(-1/2)
print(theta)
```

Observation: -0.5235987755982989
Thought: The output from the code snippet is the value of $\theta$ in radians. However, the task requires the answer to be in LaTeX format and as an exact value, not a decimal approximation. The value of $\theta$ is $\frac{5\pi}{6}$, which is the exact value of the arcsine of $-\frac{1}{2}$ within the range of the arcsine function.
Code:

```python
final_answer("\boxed{\frac{5\pi}{6}}")
```

Observation: $\boxed{\dfrac{5\pi}{6}}$ (wrong)

---

**CoT:**
To compute $\arcsin\left(-\frac{1}{2}\right)$, we need to find the angle $\theta$ such that $\sin(\theta) = -\frac{1}{2}$ and $\theta$ lies within the range of the arcsine function, which is $\left[-\frac{\pi}{2}, \frac{\pi}{2}\right]$.
1. Recall that $\sin(\theta) = -\frac{1}{2}$ corresponds to the sine of an angle in the unit circle where the y-coordinate is $-\frac{1}{2}$.
2. The sine function is negative in the third and fourth quadrants. However, since the range of the arcsine function is $\left[-\frac{\pi}{2}, \frac{\pi}{2}\right]$, we are only interested in the fourth quadrant for this problem.
3. The reference angle for $\sin(\theta) = \frac{1}{2}$ is $\frac{\pi}{6}$. Therefore, the angle in the fourth quadrant that has a sine value of $-\frac{1}{2}$ is $-\frac{\pi}{6}$.
Thus, the angle $\theta$ such that $\sin(\theta) = -\frac{1}{2}$ and $\theta \in \left[-\frac{\pi}{2}, \frac{\pi}{2}\right]$ is $-\frac{\pi}{6}$.

$\boxed{-\dfrac{\pi}{6}}$ (correct)

**Problem:** Let $f(x) = (\arccos x)^2 + (\arcsin x)^2$. Find the range of $f(x)$. All functions are in radians.

---

**Agent:**

Thought: To find the range of the function $f(x) = (\arccos x)^2 + (\arcsin x)^2$, we start by using the identity that relates the arccosine and arcsine functions:

$$\arccos x + \arcsin x = \frac{\pi}{2}.$$

Let $\theta = \arccos x$. Then $\arcsin x = \frac{\pi}{2} - \theta$. Substituting these into the function, we get:

$$f(x) = \theta^2 + \left(\frac{\pi}{2} - \theta\right)^2.$$

Expanding the expression, we have:

$$f(x) = \theta^2 + \left(\frac{\pi}{2}\right)^2 - \pi\theta + \theta^2 = 2\theta^2 - \pi\theta + \left(\frac{\pi}{2}\right)^2.$$

This is a quadratic function in terms of $\theta$. To find the range of $f(x)$, we need to determine the range of $\theta$ and then find the minimum value of the quadratic function over this range.
The range of $\theta$ is $[0, \frac{\pi}{2}]$. We can find the minimum value of the quadratic function $f(\theta) = 2\theta^2 - \pi\theta + \left(\frac{\pi}{2}\right)^2$ by completing the square or using the vertex formula for a quadratic function $ax^2 + bx + c$, which is $x = -\frac{b}{2a}$. Let's find the minimum value of $f(\theta)$ using the vertex formula.
Code:

```python
from sympy import symbols, diff, solve, pi

# Define the variable and function
theta = symbols('theta')
f_theta = 2*theta**2 - pi*theta + (pi/2)**2

# Find the derivative of the function
f_prime_theta = diff(f_theta, theta)

# Solve for the critical points
critical_points = solve(f_prime_theta, theta)
print(critical_points)
```

Observation: [π/4]

Thought: The critical point is $\theta = \frac{\pi}{4}$. We can now evaluate the function $f(\theta)$ at this point to find the minimum value of the function. Let's evaluate $f(\theta)$ at $\theta = \frac{\pi}{4}$.
Code:

```python
# Evaluate the function at the critical point
theta_critical = pi / 4
f_theta_critical = f_theta.subs(theta, theta_critical)
print(f_theta_critical)
```

Observation: $\pi^2/8$

Thought: The minimum value of the function $f(\theta)$ at $\theta = \frac{\pi}{4}$ is $\frac{\pi^2}{8}$. Since the function is a quadratic function that opens upwards (the coefficient of $\theta^2$ is positive), the minimum value is the only value in the range of $f(x)$. Therefore, the range of $f(x)$ is $\left[\frac{\pi^2}{8}, \infty\right)$.

Let's use the final_answer tool to provide the final answer in LaTeX format.
Code:

```python
final_answer("\boxed{\left[ \frac{\pi^2}{8}, \infty \right)}")
```

Observation: $\boxed{\left[\frac{\pi^2}{8}, \infty\right)}$ (wrong)

**Problem:** Let $f(x) = (\arccos x)^2 + (\arcsin x)^2$. Find the range of $f(x)$. All functions are in radians.

---

**CoT:**
To find the range of the function $f(x) = (\arccos x)^2 + (\arcsin x)^2$, we start by using the relationship between $\arccos x$ and $\arcsin x$. Recall that for any $x \in [-1, 1]$,

$$\arccos x + \arcsin x = \frac{\pi}{2}.$$

Let $\theta = \arccos x$. Then $\arcsin x = \frac{\pi}{2} - \theta$. Substituting these into the function $f(x)$, we get

$$f(x) = \theta^2 + \left(\frac{\pi}{2} - \theta\right)^2.$$

Expanding the expression, we have

$$f(x) = \theta^2 + \left(\frac{\pi}{2} - \theta\right)^2 = \theta^2 + \left(\frac{\pi}{2}\right)^2 - \pi\theta + \theta^2 = 2\theta^2 - \pi\theta + \frac{\pi^2}{4}.$$

This is a quadratic function in terms of $\theta$. To find the range of $f(x)$, we need to determine the range of $\theta$ as $x$ varies from $-1$ to $1$. Since $\theta = \arccos x$, $\theta$ ranges from $0$ to $\pi$. We need to find the minimum and maximum values of the quadratic function $g(\theta) = 2\theta^2 - \pi\theta + \frac{\pi^2}{4}$ over the interval $[0, \pi]$.

The vertex of the parabola $g(\theta) = 2\theta^2 - \pi\theta + \frac{\pi^2}{4}$ occurs at $\theta = -\frac{b}{2a} = \frac{\pi}{4}$. Evaluating $g(\theta)$ at the vertex and at the endpoints of the interval $[0, \pi]$:

1. At $\theta = 0$:

$$g(0) = 2(0)^2 - \pi(0) + \frac{\pi^2}{4} = \frac{\pi^2}{4}.$$

2. At $\theta = \pi$:

$$g(\pi) = 2(\pi)^2 - \pi(\pi) + \frac{\pi^2}{4} = 2\pi^2 - \pi^2 + \frac{\pi^2}{4} = \pi^2 + \frac{\pi^2}{4} = \frac{5\pi^2}{4}.$$

3. At $\theta = \frac{\pi}{4}$:

$$g\left(\frac{\pi}{4}\right) = 2\left(\frac{\pi}{4}\right)^2 - \pi\left(\frac{\pi}{4}\right) + \frac{\pi^2}{4} = 2 \cdot \frac{\pi^2}{16} - \frac{\pi^2}{4} + \frac{\pi^2}{4} = \frac{\pi^2}{8}.$$

The minimum value of $g(\theta)$ is $\frac{\pi^2}{8}$ and the maximum value is $\frac{5\pi^2}{4}$. Therefore, the range of $f(x)$ is

$$\left[\frac{\pi^2}{8}, \frac{5\pi^2}{4}\right].$$

Thus, the range of $f(x)$ is $\boxed{\left[\dfrac{\pi^2}{8}, \dfrac{5\pi^2}{4}\right]}$. (correct)

**Problem:** Find the smallest positive real number $C$ for which

$$\left\| \begin{pmatrix} 2 & 3 \\ 0 & -2 \end{pmatrix} \mathbf{v} \right\| \leq C \|\mathbf{v}\|$$

for all two-dimensional vectors $\mathbf{v}$.
Note that for a two-dimensional vector $\mathbf{a}$, $\|\mathbf{a}\|$ is the magnitude of $\mathbf{a}$.

---

**Small Agent distilled from trajectories without** `FTP`:
Thought: The problem is asking for the smallest positive real number C for which the inequality holds for all two-dimensional vectors v. This is a problem of finding the spectral radius of the matrix. The spectral radius of a matrix is the largest absolute value of its eigenvalues. I will first find the eigenvalues of the matrix, then find the largest absolute value of the eigenvalues, and that will be the value of C.
Code:

```
import numpy as np

# Define the matrix
A = np.array([[2, 3], [0, -2]])

# Find the eigenvalues
eigenvalues = np.linalg.eigvals(A)

# Find the largest absolute value of the eigenvalues
C = max(abs(eigenvalues))

print(C)
```

Observation:
2.0
Thought: The code has executed and the value of C has been found. Now I will return the final answer using the final_answer tool.
Code:

```
final_answer("\boxed{2}")
```

Observation: $\boxed{2}$ (wrong)

**Problem:** Find the smallest positive real number $C$ for which

$$\left\| \begin{pmatrix} 2 & 3 \\ 0 & -2 \end{pmatrix} \mathbf{v} \right\| \leq C\|\mathbf{v}\|$$

for all two-dimensional vectors $\mathbf{v}$.
Note that for a two-dimensional vector $\mathbf{a}$, $\|\mathbf{a}\|$ is the magnitude of $\mathbf{a}$.

---

**Small Agent distilled from trajectories with `FTP`:**
Thought: To find the smallest positive real number $C$ for which the inequality

$$\left\| \begin{pmatrix} 2 & 3 \\ 0 & -2 \end{pmatrix} \mathbf{v} \right\| \leq C\|\mathbf{v}\|$$

holds for all two-dimensional vectors $\mathbf{v}$, we need to analyze the norm of the matrix-vector product. The norm of the matrix-vector product can be computed as the maximum singular value of the matrix. The singular values of a matrix are the square roots of the eigenvalues of the matrix multiplied by its transpose. For a 2x2 matrix $A = \begin{pmatrix} a & b \\ c & d \end{pmatrix}$, the singular values are the square roots of the eigenvalues of $A^T A$.

Let's compute the singular values of the matrix $A = \begin{pmatrix} 2 & 3 \\ 0 & -2 \end{pmatrix}$.

Code:

```
import numpy as np

# Define the matrix
A = np.array([[2, 3], [0, -2]])

# Compute the singular values of the matrix
singular_values = np.linalg.svd(A, compute_uv=False)
print(singular_values)
```

Observation:
[4. 1.]

Thought: The singular values of the matrix $A = \begin{pmatrix} 2 & 3 \\ 0 & -2 \end{pmatrix}$ are 4 and 1. The maximum singular value is 4. Therefore, the smallest positive real number $C$ for which the inequality holds for all two-dimensional vectors $\mathbf{v}$ is the maximum singular value.
Code:

```
# The maximum singular value is the smallest C
C = max(singular_values)
final_answer(C)
```

Observation: 4 (correct)

**Example D.6: Illustrative case of first-thought prefix in factual reasoning**

**Problem:** Who founded the city where the founder of geometry lived?

---

**Small Agent distilled from trajectories with** `FTP`:

Thought: To answer this question, we need to identify the city where the founder of geometry, Euclid, lived and then determine who founded that city.

Code:

```
euclid_birthplace = web_search(query="where did Euclid live")
print(f"Euclid lived in {euclid_birthplace}.")
```

Observation:

Euclid lived in

> (...)
> "Euclid, Ohio"
> Euclid, Ohio Euclid is a city in Cuyahoga County, Ohio, United States. It is an inner ring suburb of Cleveland. As of the 2010 census, the city had a total population of 48,920. In 2009, Euclid celebrated its bicentennial. The city was named after Euclid, the Greek mathematician. The city was settled in 1796 and became a township in 1809. Euclid then became a village in 1903 and a city in 1930, during the Great Depression. The city (then a village) was the primary appealing party in the landmark case ""Village of Euclid v. Ambler Realty Co."". Euclid is located
> (...)

Thought: From the search results, we can see that Euclid, Ohio was named after the Greek mathematician Euclid. Now, we need to find out who founded the city of Euclid, Ohio. Code:

```
founder_of_euclid = web_search(query="who founded the city of
    Euclid, Ohio")
print(f"The city of Euclid, Ohio was founded by {
    founder_of_euclid}.")
```

Observation:

The city of Euclid, Ohio was founded by

> (...)
> "South Euclid, Ohio"
> treaty with the Iroquois confederation in 1796 by the Connecticut Land Company. In 1797, Moses Cleaveland named the area east of the Cuyahoga River Euclid, after the Greek mathematician and ""patron saint"" of surveyors. Euclid Township was officially formed in 1809. In 1828, Euclid Township was divided into nine districts, with South Euclid becoming district two.
> (...)

Thought: The search results indicate that the city of Euclid, Ohio was settled in 1796 by Moses Cleaveland, who was trained in mathematics and named the township after the Greek mathematician Euclid. Therefore, Moses Cleaveland founded the city where the founder of geometry, Euclid, lived. Code:

```
final_answer("Moses Cleaveland")
```

Observation:

Moses Cleaveland (wrong)

**Problem:** Who founded the city where the founder of geometry lived?

**Small Agent distilled from trajectories without `FTP`:**

Thought: I need to find out who founded the city where the founder of geometry lived. I will first find out who the founder of geometry is, then find out where he lived, and finally find out who founded the city where he lived. I will use the tool 'web_search' to get this information.
Code:

```
founder_of_geometry = web_search(query="founder of geometry")
print(f"Founder of geometry: {founder_of_geometry}")
```

Observation:
Founder of geometry:

> (...) Euclid
> Euclid Euclid (; – ""Eukleídēs"", ; fl. 300 BC), sometimes given the name Euclid of Alexandria to distinguish him from Euclides of Megara, was a Greek mathematician, often referred to as the ""founder of geometry"" or the ""father of geometry"". He was active in Alexandria during the reign of Ptolemy I (323–283 BC). His ""Elements"" is (...)

Thought: From the search results, I can see that the founder of geometry is Euclid. Now I need to find out where he lived and who founded the city where he lived. I will use the tool 'web_search' again to get this information. Code:

```
euclid_lived = web_search(query="where did Euclid live")
print(f"Where did Euclid live: {euclid_lived}")
```

Observation:
Where did Euclid live:

> (...) Euclid
> A detailed biography of Euclid is given by Arabian authors, mentioning, for example, a birth town of Tyre. This biography is generally believed to be fictitious. If he came from Alexandria, he would have known the Serapeum of Alexandria, and the Library of Alexandria, and may have worked there during his time. (...)

Thought: From the search results, I can see that Euclid lived in Alexandria, Egypt. Now I need to find out who founded the city where Euclid lived. I will use the tool 'web_search' again to get this information. Code:

```
founder_of_alexandria = web_search(query="who founded Alexandria"
    )
print(f"Who founded Alexandria: {founder_of_alexandria}")
```

Observation:
Who founded Alexandria:

> (...) Alexandria
> Alexandria is believed to have been founded by Alexander the Great in April 331 BC as (""Alexandreia""). Alexander's chief architect for the project was Dinocrates. Alexandria was intended to supersede Naucratis as a Hellenistic center in Egypt, (...)

Thought: From the search results, I can see that Alexandria was founded by Alexander the Great. Therefore, the founder of the city where the founder of geometry lived is Alexander the Great. I will now provide the final answer using the 'final_answer' tool. Code:

```
final_answer("Alexander the Great")
```

Observation:
Alexander the Great (correct)

