# OpenReview forum: "Distilling LLM Agent into Small Models with Retrieval and Code Tools"
_NeurIPS.cc/2025/Conference — NeurIPS 2025 spotlight_

### Official Review · Reviewer_yrcR · 2025-06-06

**Clarity:** 3
**Significance:** 3
**Originality:** 3
**Rating:** 5
**Confidence:** 3

**Summary:**

The authors introduce Agent Distillation, a training framework that teaches small language models to learn the reason-act-observe behavior of large tool-using LLM agents rather than just learning their chain-of-thought text. They introduce two techniques: a first-thought prefix that elicits teacher trajectories without extra fine-tuning, and self-consistent action generation that samples multiple tool-using trajectories at inference and chooses the one that yields a valid, consistent answer. Across eight factual and mathematical benchmarks, the distilled agents outperform standard CoT-distilled models.

**Questions:**

- One common pitfall of distillation is overfitting to teacher rollouts.  Did the author observe any overfitting in their experiments and how to prevent them?
- How many rollouts did the authors collect from teacher per prompt?

**Ethical Concerns:**

["NO or VERY MINOR ethics concerns only"]

**Final Justification:**

Issues resolved

**Limitations:**

Yes

**Quality:**

3

**Strengths And Weaknesses:**

Strength
- The experiments are extensive and insightful, spanning across various model sizes (0.5B - 7B) and datasets (e.g., MATH, HotpotQA, AIME, Bamboogle). The authors also provide qualitative examples and failure case analysis.
- The agent distillation idea is well-motivated and well-executed


Weakness
- The training method is a pretty straightforward extension of existing distillation methods and may slightly lack novelty. But a simple method that works well is a reasonable contribution.

---

> ### Author Rebuttal · Authors · 2025-07-31
>
> We sincerely thank the reviewer for the thoughtful comments and positive assessment. We are glad that you found (1) the agent distillation idea well-motivated and well-executed, and (2) appreciated the breadth of our experiments, including the diverse benchmarks and model sizes. Below, we address the questions and concerns in detail.
>
> ---
>
> > W1. The training method is a pretty straightforward extension of existing distillation methods and may slightly lack novelty. But a simple method that works well is a reasonable contribution.
>
> We appreciate the recognition that a simple method with strong empirical results can be a valuable contribution. **Although our framework builds on distillation, it introduces two novel components tailored for agent distillation:**
>
> - First-Thought Prefix (FTP): Unlike standard prompting, FTP prepends a first thought generated via CoT prompting to guide the agent's reasoning, improving performance on reasoning-intensive tasks for both LLM (Table 4) and distilled agents (Table 2).
> - Self-Consistent Action Generation (SAG): his extends self-consistency to agent settings by sampling diverse action trajectories and selecting valid, executable ones based on environment feedback, thereby enhancing robustness and action quality.
>
> These components, to our knowledge, have not been explored in the context of tool-using agent distillation.
>
> ---
>
> > Q1. One common pitfall of distillation is overfitting to teacher rollouts. Did the author observe any overfitting in their experiments and how to prevent them?
>
> To assess generalization, we evaluated the distilled agents on six out-of-distribution tasks, some of which are more challenging than the training tasks (e.g., AIME vs. MATH, MuSiQue (3-hop) vs HotpotQA (2-hop)). The consistent performance improvements over CoT-distilled baselines on those tasks suggest that **overfitting is limited**.
> We also used LoRA-based fine-tuning, which is known to reduce overfitting compared with full fine-tuning [1].
>
> [1] Biderman et al., LoRA Learns Less and Forgets Less, TMLR 2024
>
> ---
>
> > Q2. How many rollouts did the authors collect from teacher per prompt?
>
> **We collected one teacher rollout per prompt for training.** As suggested in prior work on CoT distillation [1], using multiple rollouts may further improve performance, which we plan to explore in future extensions.
>
> [1] Ho et al., Large Language Models Are Reasoning Teachers, ACL 2023

---

### Official Review · Reviewer_NTqR · 2025-06-23

**Clarity:** 4
**Significance:** 4
**Originality:** 4
**Rating:** 5
**Confidence:** 4

**Summary:**

The paper introduces Agent Distillation, a framework for distilling agentic behavior into small LLMs. Besides naive distillation, the authors provide two precious insights that help boost results further. They train a family of models ranging from 0.5B to 3B parameters on 2 datasets, showing generalization capabilities on the other 6.

**Questions:**

- Could the authors provide additional experiments involving llama, phi, mistral, or other alternatives? It would bring more certainty about the Agent Distillation's strengths.
- Have the authors thought of any ways of mitigating the downsides of the first-thought prefix (i.e., reduction of retrieval calls)?
- Could the authors further discuss whether code execution during SAG is safe? Given the stochasticity of decoding, is there a risk of producing and executing dangerous or irreversible commands from the long tail of the model's distribution?

**Ethical Concerns:**

["NO or VERY MINOR ethics concerns only"]

**Final Justification:**

The presented method is clearly described and shows consistent improvements over baselines. I appreciate the additional evaluations shared during the discussion phase, which support the generality of the approach across different model families. The authors also included a complete codebase that seems clean and well-integrated with Hugging Face, which adds to the paper's practical value.

That said, I suggest the authors take into account the feedback from the other reviewers as well, especially regarding novelty and positioning relative to related work, even if I personally found the paper convincing. I recommend acceptance.

**Limitations:**

yes

**Quality:**

3

**Strengths And Weaknesses:**

## Strengths
 - Clear paper, full code: To me, the paper reads very clearly, and everything seems in place. I had a look at the code as well and it seems clear as well.
 - Well-motivated tricks with ablations: Both First-Thought Prefix and Self-Consistent Action Generation are simple but powerful ideas. The authors made sure to conduct enough experiments to show their strengths and potential downsides.
 - Cross-domain evaluation showcases generalization: The method is evaluated on eight diverse benchmarks covering both factual and mathematical reasoning, including out-of-distribution generalization. Small models consistently match larger CoT-based ones.
 - Small models and training set: Agent behavior is distilled into models as small as 0.5B parameters using only 1k-2k teacher trajectories.

## Weaknesses
 - Evaluation on single model family: Even though it is acknowledged as a limitation, all experiments were conducted on Qwen models.
 - SAG poses some safety risks: Executing code sampled from a stochastic policy could lead to irreversible or unsafe operations.

---

> ### Author Rebuttal · Authors · 2025-07-31
>
> We sincerely thank the reviewer for the feedback and their appreciation of our work on multiple aspects.
> We appreciate the reviewer's recognition of (1) our clear writing, (2) our complete code release, (3) thorough ablations, (4) simple yet effective techniques, and (5) strong generalization across diverse benchmarks. Below, we address each concern.
>
> ---
>
> > W1. Evaluation on single model family: Even though it is acknowledged as a limitation, all experiments were conducted on Qwen models
>
> > Q1. Could the authors provide additional experiments involving llama, phi, mistral, or other alternatives? It would bring more certainty about the Agent Distillation's strengths.
>
> Thank you for the thoughtful comment. We agree that evaluating our method across multiple model families would strengthen the generality of our findings. **We plan to extend our evaluation to other small models**, such as LLaMA-3.2-1B-Instruct and Phi-3-mini-instruct, in a future revision.
>
> ---
>
> > W2. SAG poses some safety risks: Executing code sampled from a stochastic policy could lead to irreversible or unsafe operations.
>
> > Q3. Could the authors further discuss whether code execution during SAG is safe? Given the stochasticity of decoding, is there a risk of producing and executing dangerous or irreversible commands from the long tail of the model's distribution?
>
> We appreciate the reviewer raising this important concern. In practice, **such risks can be mitigated through sandboxing mechanisms (e.g., Docker, E2B Sandbox)**, which are widely adopted in agent systems such as OpenHands [1]. Additionally, incorporating a safety guard model to verify code safety prior to execution is another viable future direction. We have clarified this in the discussion section.
>
> [1] Wang et al., OpenHands: An Open Platform for AI Software Developers as Generalist Agents, ICLR 2025
>
> ---
>
> > Q2. Have the authors thought of any ways of mitigating the downsides of the first-thought prefix (i.e., reduction of retrieval calls)?
>
> One possible solution is to **apply reinforcement learning after distillation** to encourage effective retrieval behaviors in students. While we did not pursue this direction in this work, we agree it is a promising avenue for future research in training small agents.

---

> > ### Comment · Reviewer_NTqR · 2025-08-01
> >
> > Thank you for your rebuttal.
> >
> > Did the authors manage to conduct at least some preliminary experiments with other models?
> > Is extending the evaluation to other small models a complex process?
> >
> > I think that providing more details on this would help strengthen the submission.

---

> ### Author Response · Authors · 2025-08-01
>
> We sincerely thank the reviewer for the thoughtful and prompt response.
>
> We fully agree that evaluation on additional small models is important, and we apologize for not including at least preliminary results in our rebuttal.
> While the process itself is not complex, our experiments have been delayed due to limited GPU availability.
> **We are currently running evaluations on other small models and will share preliminary results within the next two days.**
>
> Thank you again for your continued engagement and constructive feedback.

---

> ### Author Response · Authors · 2025-08-03
>
> Dear Reviewer NTqR,
>
> We are pleased to share our preliminary evaluation results for **Llama-3.2-1B-Instruct**:
>
> | Method          | HotpotQA | MATH500 | Musique | Bamboogle | 2Wiki | GSM8K | AIME | Olymath |  Average |
> | --------------- | -------: | ------: | ------: | --------: | ----: | ----: | ---: | ------: | -------: |
> | CoT Prompting       |     13.2 |    28.8 |     1.2 |      14.4 |     8 |    19 | 1.11 |     2.5 | 11.03 |
> | CoT Distill  |     18.2 |    25.6 |     2.6 |      **25.6** |    19 |  13.8 | 1.11 |       2 | 13.50 |
> | Agent Distill       |       36 |    34.6 |     2.6 |      11.2 |  26.4 |  40.4 | 1.11 |       2 | 19.29 |
> | Agent Distill (FTP)        |     37.6 |    32.8 |     **3.6** |      24.0 |  **30.8** |    45 | 1.11 |     1.5 | 22.05 |
> | Agent Distill (FTP + SCA)  |     **40.6** |      **40** |     3.2 |      23.2 |    30 |  **47.8** | 1.11 |       **3** | **23.61** |
>
> These results mirror the trends we observed with Qwen-2.5 models: our distilled agent with both FTP and SAG **outperforms** the distilled CoT baseline by **over 10 points** on average. **The performance gains from applying FTP and SAG are similarly consistent**, demonstrating the effectiveness of our methods across different model families.
>
> We have included these findings into the revised manuscript. We are also running evaluations on additional small models, such as Phi-4-mini-instruct (3.8B), and will report those results during the discussion phase.
>
> Please let us know if you have any further questions or comments. We would be happy to address them. Thank you again for your time and valuable feedback.

---

> ### Author Response · Authors · 2025-08-06
>
> Dear Reviewer NTqR,
>
> We would like to share additional evaluation results on **Phi-4-mini-instruct (3.8B)**, which further support the generality of our proposed method:
>
> | Method                  | HotpotQA | MATH500 | Musique | Bamboogle | 2Wiki | GSM8K | AIME | Olymath | Average  |
> |-----|------:|-----:|-----:|-----:|------:|------:|------:|------:|------:|
> | CoT Prompting           | 24.2     | 53.8    | 6.0     | **38.4**      | 24.2  | 49.6  | 5.56 | 4.5     | 25.78    |
> | CoT Distill             | 24.4     | 63.2    | 5.8     | 33.6      | 24.8  | 54.8  | 6.67 | 7.0     | 27.53    |
> | Agent Distill           | **48.2**     | 52.4    | 8.8     | 27.2      | 33.6  | 69.4  | 5.56 | 6.0     | 31.40    |
> | Agent Distill (FTP)     | 45.2     | 60.0    | 7.2     | 34.4      | 39.2  | 71.2  | 10.0 | **7.5**     | 34.34    |
> | Agent Distill (FTP+SCA) | 47.0     | **65.6**    | **9.6**     | 32.0      | **41.0**  | **73.0**  | **11.11** | 7.0     | **35.79** |
>
> These results are consistent with the trends we observed on Qwen2.5 and Llama3 models, further confirming that our approach is effective across multiple model families. We have updated the revised manuscript to reflect these findings.
>
> If you have any further concerns or questions, please feel free to share them during the discussion phase. We would be more than happy to engage.
>
> Best regards,
>
> The Authors

---

> > ### Comment · Reviewer_NTqR · 2025-08-07
> >
> > Thank you for sharing the new results and for taking the time to run these extra evaluations. It's great to see that the improvements hold across different small models. I maintain my positive score.

---

> > > ### Author Response · Authors · 2025-08-08
> > >
> > > Dear Reviewer NTqR,
> > >
> > > Thank you for your positive follow-up and for recognizing the consistency of our improvements across different small models. Your continued support is highly encouraging, and we will incorporate these results into the final version. Thank you again for your time and effort in reviewing our work.
> > >
> > > Best regards,
> > >
> > > The Authors

---

### Official Review · Reviewer_EVtf · 2025-07-01

**Clarity:** 3
**Significance:** 1
**Originality:** 1
**Rating:** 4
**Confidence:** 4

**Summary:**

The paper introduces the Agent Distillation framework that distills tool-calling skills (e.g., search, code execution, etc.) from large-scale LLMs to smaller ones. The method proposes two key innovations: a new "first-thought prefix" (FTP) prompting scheme for enhancing the clarity of the teacher trajectories, along with the "self-consistent action generation" (SAG), which is a self-consistency method applied to the results of the code execution.

**Questions:**

See weaknesses.

**Ethical Concerns:**

["NO or VERY MINOR ethics concerns only"]

**Final Justification:**

The provided commentary on the difference between the existing work and underscoring the need for tools affected my decision to raise the score.

**Limitations:**

Yes

**Quality:**

2

**Strengths And Weaknesses:**

Strengths:
 - The paper is written clearly and is easy to follow.
 - The paper presents evaluations on multiple benchmarks with extensive analysis.

Weaknesses:
 - No particular novelty in the method -- same distillation loss, but the CoT trajectory is substituted with the agentic one. The prompting and self-consistency in generating code indeed enhance the quality of the trajectories, but have already been studied in the literature, e.g. [6].
- [1] uses a large-scale LLM to collect a dataset of reasoning chains and tool calls and then applies DPO to train a smaller LLM. The proposed method should be compared to this baseline. Overall, the related works miss a lot of concurrent work, e.g., [4].
 - The main benefit over the original CoT distillation comes from using tools, not the distillation method itself, as can be seen by the higher average score of the teacher model when allowed to use tools, as well as a <= 3% difference in scores between distillation w and w/o FTP+SAG. Thus, it would be fair to compare with some of the tool-learning methods, like TinyAgent [2], ToolkenGPT, and others [3, 4].

[1]. https://arxiv.org/abs/2410.18890
[2]. https://aclanthology.org/2024.emnlp-demo.9.pdf
[3]. https://arxiv.org/abs/2302.04761
[4]. https://arxiv.org/abs/2502.05867
[5]. https://arxiv.org/abs/2407.00121
[6]. https://arxiv.org/abs/2309.17272

---

> ### Author Rebuttal · Authors · 2025-07-31
>
> We thank the reviewer for the feedback and appreciate that they found our work (1) clearly written and (2) evaluated on multiple benchmarks with extensive analysis.
> We believe some concerns may stem from a lack of clarity about certain aspects of our contributions. In the following, we provide explanation to clarify our novelty and distinguish our work from prior works.
>
> ---
>
> > W1. No particular novelty in the method -- same distillation loss, but the CoT trajectory is substituted with the agentic one. The prompting and self-consistency in generating code indeed enhance the quality of the trajectories, but have already been studied in the literature, e.g. [6].
>
> The contribution of our work is not based solely on methodological novelty. As acknowledged by other reviewers, our work provides well-motivated methods (NTqR, yrcR), comprehensive and extensive evaluations (qN16, NTqR, yrcR), and a valuable contribution to the field (qN16).
> We also argue that **our method does introduce meaningful contributions beyond simply applying existing techniques**, as detailed below:
>
> - First, the First-Thought Prefix (FTP) is not a naive reuse of existing prompting methods. We identified a specific failure mode of LLM agents in reasoning tasks and proposed a novel solution: prepending an initial "first thought" derived from CoT prompting to better steer the agent's reasoning process. This approach results in improved performance for both the teacher (Table 4) and student models (Table 2). To our knowledge, **no prior work has applied similar approach to agentic reasoning or explored its advantages for distillation.**
> - Second, our Self-Consistent Action Generation (SAG) adapts the idea of self-consistency decoding to the agentic setting. Unlike prior methods that rely on ground-truth answers [1] or external LLM judges [2], our approach uses majority voting over sampled actions, based solely on observations from the environment. **This adaptation and its demonstrated effectiveness (Table 2) have not been explored in previous work, highlighting its contribution.**
>
> [1] Wang et al., Self-Consistency Improves Chain of Thought Reasoning in Language Models, ICLR 2023
>
> [2] Chen et al., Universal Self-Consistency for Large Language Model Generation
>
> ---
>
> > W2. [1] uses a large-scale LLM to collect a dataset of reasoning chains and tool calls and then applies DPO to train a smaller LLM. The proposed method should be compared to this baseline. Overall, the related works miss a lot of concurrent work, e.g., [4].
>
> We clarify that while our work and Manduzio et al. (2024) [1] share the high-level goal of training small models using trajectories from LLM agents, **their objectives, tool settings, training methodology, and evaluation scope differ substantially**, as summarized below.
>
> | Aspect           | **Our Work**                                      | **Manduzio et al. (2024) [1]**                                        |
> |---|---|---|
> | **Objective**    | Teach small models *interactive agent behavior* with tools            | Teach small models *function-calling ability* on fixed tasks     |
> | **Tool Use**     | Open-ended tools (retrieval, Python code) in math and factual reasoning          | Pre-defined functions (e.g., `add()`, `Stop()`) in simple logic/math tasks    |
> | **Training**     | Supervised learning on reason-act-observe trajectories                | DPO (preference-based learning from correct/incorrect completions) |
> | **Evaluation**   | 8 challenging reasoning-intensive tasks (HotpotQA, MATH, AIME, etc.)                         | 2 simple tasks (GSM8K, small FOL benchmark)                           |
> | **Target Small Model Sizes**   | 0.5B, 1.5B, 3B, and 7B    | 7B only         |
>
> Due to these differences, we believe a direct comparison would not be meaningful. We have cited this work and clarified its distinction in the revision.
>
> ---
>
> > W3. The main benefit over the original CoT distillation comes from using tools, not the distillation method itself, as can be seen by the higher average score of the teacher model when allowed to use tools, as well as a <= 3% difference in scores between distillation w and w/o FTP+SAG. Thus, it would be fair to compare with some of the tool-learning methods, like TinyAgent [2], ToolkenGPT, and others [3, 4].
>
> The teacher’s performance gain indeed stems from its ability to use tools. However, we argue that this fact **underscores the necessity and value of our agent distillation method.**
>
> Our central hypothesis is that **a student model cannot acquire tool-use capabilities through simple prompting alone**. As clearly shown in Table 2, directly prompting the student to use tools yields minimal or even negative gains, where 0.5B model achieves nearly zero accuracy.
>
> This demonstrates that **only through distillation of the teacher's agent trajectories can tool-use ability be effectively transferred to the student.** In other words, agent distillation is the mechanism that unlocks the potential of tools for small language models (sLMs).
>
> Furthermore, while the 2-3 point absolute gain from FTP+SAG may appear modest, it is highly significant for sLMs, corresponding to **their relative gains of 13.83%, 8.87%, and 8.93% for 0.5B, 1.5B and 3B models, respectively.**
>
> Therefore, our work should be viewed not as a simple tool-learning method, but as a study that **identifies the limits of CoT distillation and pushes the capability boundaries of sLMs by the agent distillation paradigm.**

---

> > ### Comment · Reviewer_EVtf · 2025-08-05
> >
> > I thank the authors for their response and acknowledge the extensive evaluations, raising my score to 4.

---

> > > ### Author Response · Authors · 2025-08-06
> > >
> > > Dear Reviewer EVtf,
> > >
> > > Thank you for taking the time to read our rebuttal and for updating your score accordingly. We have included comparisons with the works you suggested in our revision, which we believe help clarify the positioning of our work. We truly appreciate your time and effort in helping us improve the paper.
> > >
> > > Best regards,
> > >
> > > The Authors

---

### Official Review · Reviewer_qN16 · 2025-07-02

**Clarity:** 3
**Significance:** 2
**Originality:** 2
**Rating:** 4
**Confidence:** 4

**Summary:**

This paper introduces Agent Distillation, a method to transfer the reasoning and tool-use capabilities of LLMs to sLMs. The approach leverages LLM-generated trajectories of reason-act-observe cycles, incorporating two key techniques: First-Thought Prefix (FTP) to enhance the quality of the teacher LLM's trajectories, and Self-Consistent Action Generation (SAG) to improve the robustness of the student sLMs at test time.  Experiments on factual and math reasoning tasks show that agent-distilled sLMs outperform larger traditional CoT-distilled models.

**Questions:**

1. In describing Self-Consistent Action Generation (SAG), you state that you sample with a "high temperature" to encourage diversity, yet the value used is 0.4. In many contexts, this is considered a relatively low temperature that promotes less randomness. Could you clarify this choice and provide any ablation studies you performed on the effect of temperature on SAG's performance?
2. Could you provide more detail on the evaluation protocol? Specifically, were the reported scores, especially for benchmarks with small size like AIME, averaged over multiple runs with different random seeds to ensure statistical significance?
3. For the student models, you chose to use LoRA for fine-tuning. Given the relatively small size of both the models and the datasets, what was the rationale for using LoRA instead of full fine-tuning? Did you experiment with full fine-tuning, and if so, how did the results compare?
4. An interesting trend appears on the MATH500 benchmark, where standard CoT prompting outperforms your distillation method for the 3B and 7B models, reversing the trend seen with the 0.5B and 1.5B models. Do you have a hypothesis for this scale-dependent performance difference?
5. The methodology states that you "filter out any sequences that result in parsing or execution errors" (lines 160-161). Logically, this filtering should result in a zero error rate for the final selected action. However, the results in Figure 6 show that SAG still produces a non-zero error rate. Could you clarify why the error rate is non-zero, or how is it calculated?

**Ethical Concerns:**

["NO or VERY MINOR ethics concerns only"]

**Final Justification:**

The new AIME, multiple runs and full fintuning experiment have been further conducted to validate the effectiveness of the paper method. The novelty problem although still build on some existing methods, but has also certain difference.

**Limitations:**

● The effectiveness of the "first-thought prefix" (FTP) is only validated on a single 32B instruction-tuned teacher model. It is not clear whether this strategy is broadly applicable to other models or families, particularly Large Reasoning Models (LRMs) like DeepSeek-R1 or OpenAI-o1.
● The paper's core technical contributions, FTP and SAG, are better described as effective applications of existing ideas (prompt engineering and self-consistency decoding, respectively) rather than fundamentally new methods. This limits the paper's originality.
● The evaluation lacks details on its statistical robustness. The authors do not specify if results were averaged over multiple runs with different random seeds. This is a significant limitation, particularly for benchmarks with small test sets like AIME, where performance can have high variance and single-run scores are not reliable indicators of true performance.

**Quality:**

2

**Strengths And Weaknesses:**

Strengths:
1. Agent Distillation offers a practical and effective pathway for transferring the agentic capabilities of LLMs into smaller, more efficient sLMs. This presents a valuable contribution towards developing powerful on-device AI agents that can execute complex tasks without the high computational costs associated with LLMs.
2. The paper is well-structured and the experiments are comprehensive. The authors evaluate their method on a diverse set of eight benchmarks, covering both in-domain and out-of-domain generalization for factual and mathematical reasoning.

Weaknesses:
1. While their application is effective, the core methods are derivative of existing work. The first-thought prefix (FTP) is a well-executed form of prompt engineering, designed to elicit a specific reasoning format. The self-consistent action generation (SAG) is a direct application of the well-established Self-Consistency decoding strategy.
2. The paper proposes the first-thought prefix (FTP) to improve teacher quality but only validates it on a single model (Qwen2.5-32B-Instruct). It remains a question whether this specific prompting strategy is effective for other, more powerful reasoning models like DeepSeek-R1 or the OpenAI-o1, which may have different internal reasoning structures.
3. The experimental analysis, while broad, lacks depth in key areas. The individual contributions of FTP and SAG are not clearly disentangled, making it difficult to quantify their separate impacts on the final performance. Key hyperparameters, such as the sampling temperature for SAG, are presented without ablation studies to justify their chosen values.

---

> ### Author Rebuttal · Authors · 2025-07-31
>
> We thank the reviewer for their valuable and insightful feedback.
> We are glad to hear that the reviewer found our work (1) offers a practical and effective pathway for transferring the agentic capabilities, (2) presents a valuable contribution towards developing powerful on-device agents, (3) is well-structured and the experiments are comprehensive.
>
> To address your concerns, we have clarified our contribution in terms of methods, run additional suggested experiments regarding ablation studies on SAG and multiple runs on AIME.
>
> ---
>
> > W1. While their application is effective, the core methods are derivative of existing work. The first-thought prefix (FTP) is a well-executed form of prompt engineering, designed to elicit a specific reasoning format. The self-consistent action generation (SAG) is a direct application of the well-established Self-Consistency decoding strategy.
>
> We acknowledge that both the First-Thought Prefix (FTP) and Self-Consistent Action Generation (SAG) are inspired by prior work. However, we would like to clarify that our contribution lies in **tailoring and validating these ideas in the novel context of agent distillation.**
>
> - FTP addresses the performance drop of language model agents in reasoning tasks. We find **prepending the first-thought from CoT in generating agent trajectories improves the performance of LLM agent by +5.07 (Table 4) as well as downstream performance of distilled agent (1.5B) by +1.05 (Table 2).** To our knowledge, this problem has not been identified or addressed in prior work.
> - For SAG, while it is motivated by self-consistency decoding, **our work is the first to adapt and evaluate it in the context of action generation for interactive language agents.** Unlike traditional uses of self-consistency that rely on access to ground-truth answers or additional LLM as a judge [1], we leverage majority voting over **observations from the environment** without external supervision. Importantly, this method remains effective even for the smallest 0.5B model evidenced by +1.91 performance improvement in average for 8 tasks (Table 2).
>
> We hope this clarifies that our contributions extend beyond applying existing ideas.
>
> [1] Chen et al., Universal Self-Consistency for Large Language Model Generation
>
> ---
>
> > W2. The paper proposes the first-thought prefix (FTP) to improve teacher quality but only validates it on a single model (Qwen2.5-32B-Instruct). It remains a question whether this specific prompting strategy is effective for other, more powerful reasoning models like DeepSeek-R1 or the OpenAI-o1, which may have different internal reasoning structures.
>
> Although we have not yet evaluated FTP on DeepSeek-R1 or OpenAI models because of computational-budget constraints, we agree that testing a reasoning model as teacher would strengthen our claim. We therefore experimented with a feasible open-source reasoning model, QwQ-32B. However, we observed that QwQ-32B solved most problems entirely inside its thinking steps without tools, so meaningful agentic traces are hard to be collected. This result suggests that strong reasoning models may bypass tool use, limiting straight-forward agent distillation. We have included this discussion to the revision and leave using reasoning models as the teacher for future work.
>
> ---
>
> > W3. The experimental analysis, while broad, lacks depth in key areas. The individual contributions of FTP and SAG are not clearly disentangled, making it difficult to quantify their separate impacts on the final performance. Key hyperparameters, such as the sampling temperature for SAG, are presented without ablation studies to justify their chosen values.
>
> We already provide an ablation of FTP alone and FTP+SAG in Table 2, showing that applying each method progressively improves performance. To further isolate the effect of SAG, we conducted additional experiments using SAG alone on the 0.5B, 1.5B, and 3B models. As shown in the table below, SAG alone improves average performance by up to +2.2 points (1.5B) and consistently yields benefits across all scales.
>
> |        | HotpotQA  | MATH500  | MuSiQue | Bamboogle | 2Wiki | GSM8K | AIME | OlymMAth | Avg    |
> |--------|-----|-----|----|----|----|----|----|----|--------|
> | D (3B)      | 48.4  |54 | 13  | 37.6 | 37.4 | 64.2 | 6.67 | 7.5  | 33.60   |
> | D+SAG (3B)  | 48.6 | 57.4 | 13 | 36 |37.4| 65.6 | 6.67 | 10  | **34.33**   |
> | D (1.5B)      | 43  |46.8 | 9  |27.2|35.6|54.8|1.1 | 7  |28.06   |
> | D+SAG (1.5B)  |43.8 |49.8 |11.6|31.2|36.6| 58 |7.8 | 3.5  | **30.29**   |
> | D (0.5B)    |34.6 |30.4 | 7  |17.6|28.8|31.2|3.3 | 1  |19.24   |
> | D+SAG (0.5B) | 34  |33.8 | 8.2|13.6| 33 | 33 |4.4 | 0  | **20.01**   |
> * D denotes the distilled agent.
>
> We provide the ablation on sampling temperature in our response to Q1.
>
> ---
>
> > Q1. In describing Self-Consistent Action Generation (SAG), you state that you sample with a "high temperature" to encourage diversity, yet the value used is 0.4. In many contexts, this is considered a relatively low temperature that promotes less randomness. Could you clarify this choice and provide any ablation studies you performed on the effect of temperature on SAG's performance?
>
> Thank you for pointing this out. We acknowledge the term "high temperature" in Line 159 can be misleading. Our intent was to indicate a shift from deterministic decoding (temperature = 0) to sample diverse actions. We chose 0.4, as performance did not vary significantly across temperatures in SAG.
>
> As suggested by the reviewer, we conducted a temperature ablation study on SAG for math tasks using the 1.5B model with temperatures 0.2, 0.4, 0.6, 0.8, and 1.0. The results are shown below and have been included in the revision.
>
> | Temperature | MATH500 | GSM-Hard | AIME | OlymMATH | Avg (Math) |
> |---|---|---|---|---|---|
> | 0.2         | 48.0    | 60.2     | **7.78** | 3.5       | 29.87       |
> | 0.4         | 50.6    | 60.6     | 6.67 | **4.5**       | 30.59   |
> | 0.6         | 50.8    | 61.8     | 4.44 | **4.5**       | 30.39       |
> | 0.8         | **52.4**    | 61.8     | 4.44 | 3.5       | 30.54       |
> | 1.0         | 51.0    | **63.8**     | 6.67 | 3.5       | **31.24**       |
>
> ---
>
> > Q2. Could you provide more detail on the evaluation protocol? Specifically, were the reported scores, especially for benchmarks with small size like AIME, averaged over multiple runs with different random seeds to ensure statistical significance?
>
> We report all experimental results with a single run due to computational budget constraints, as noted in Checklist item 7. We have clarified this detail in the experimental setup section of the revision.
>
> To assess variance on smaller benchmarks, we additionally ran AIME with 5 different seeds for each model size with the FTP and SAG methods. The results below report the average and standard deviation:
>
> | Model       | Avg  | Std  |
> |-------------|------|------|
> | 0.5B  | 2.00 | 0.93 |
> | 1.5B  | 6.23 | 0.61 |
> | 3B    |14.44 | 1.36 |
>
> These results show that the distilled agents yield **consistent performance on AIME** with relatively low standard deviation across model sizes. We have also included these statistics in the revised manuscript to enhance statistical rigor on the small benchmark.
>
> ---
>
> > Q3. For the student models, you chose to use LoRA for fine-tuning. Given the relatively small size of both the models and the datasets, what was the rationale for using LoRA instead of full fine-tuning? Did you experiment with full fine-tuning, and if so, how did the results compare?
>
> The primary rationale for using LoRA was its efficiency in terms of both reduced training cost and ease of deployment. Although we did not experiment with full fine-tuning, we believe this is a valuable direction and plan to include it in the future revision.
>
> ---
>
> > Q4. An interesting trend appears on the MATH500 benchmark, where standard CoT prompting outperforms your distillation method for the 3B and 7B models, reversing the trend seen with the 0.5B and 1.5B models. Do you have a hypothesis for this scale-dependent performance difference?
>
> Thank you for this insightful observation. Our hypothesis is as follows:
>
> - Smaller models (0.5B, 1.5B) benefit greatly from tool use, as it compensates for their limited arithmetic ability.
> - Larger models (3B, 7B) have stronger internal computation skills, making tool use less beneficial on simpler benchmarks like MATH500.
>
> However, **agent distillation remains effective for 3B and 7B models.** As shown in Figure 3, the 3B distilled agent performs better on harder MATH500 problems (level 5) with the agentic method.
> Moreover, on benchmarks involving more complex calculations (e.g., GSM-Hard, OlymMATH), the performance gains from agent distillation are consistent across 3B and 7B models. We have clarified this task-dependence in the experimental results section.
>
> ---
>
> > Q5. The methodology states that you "filter out any sequences that result in parsing or execution errors" (lines 160-161). Logically, this filtering should result in a zero error rate for the final selected action. However, the results in Figure 6 show that SAG still produces a non-zero error rate. Could you clarify why the error rate is non-zero, or how is it calculated?
>
> We thank the reviewer for highlighting this. The filtering rule, discarding samples with parsing or execution errors, **applies only when at least one valid action exists.**
>
> When all generated actions fail, we retain randomly-selected one failed action and feed its error message back as an observation, allowing the model to self-correct. The non-zero error rate in Figure 6 reflects these cases. We have clarified this in the method section to avoid confusion.

---

> ### Author Response · Authors · 2025-08-06
>
> Dear Reviewer qN16,
>
> We would like to follow up regarding **Q3**, as we have now conducted the **full fine-tuning (FT) experiment** that we were unable to include in the rebuttal due to resource constraints. Below, we share the preliminary results on Qwen2.5-1.5B-Instruct:
>
> | Method (FTP)                     | HotpotQA | MATH500 | Musique | Bamboogle | 2Wiki | GSM8K | AIME | Olymath | Average  |
> |----------:|---------:|---------:|---------:|-----------:|-------:|-------:|------:|---------:|----------:|
> | Agent Distill (LoRA)           | **43.6**     | **46.4**    | **8.0**     | **30.4**      | 32.6  | **60.6**  | **7.78** | 3.5     | **29.11**    |
> | Agent Distill (Full FT)        | 40.6     | 45.2    | 6.2     | 20.0      | **35.0**  | 52.0  | 4.44 | **6.5**     | 26.24    |
>
> As shown, full fine-tuning does not yield better performance on average compared to LoRA-based fine-tuning. While further hyperparameter tuning may help, these results suggest that full fine-tuning could hurt generalization possibly due to overfitting. We have incorporated this result and analysis into the revised manuscript.
>
> Thank you again for your time and thoughtful review. We hope that our rebuttal has addressed most of the concerns you raised. If there are any remaining questions, concerns, or clarifications you would like us to address, we would be sincerely grateful for the opportunity to discuss them during the discussion phase.
>
> Best regards,
>
> The Authors

---

> > ### Comment · Reviewer_qN16 · 2025-08-08
> > **Official Comment by Reviewer qN16**
> >
> > Thanks for the detailed rebuttal and added experiment. It has addressed most of my concerns, I will increase the score to 4, and the new experiment should be added to the final version.

---

### Note · Authors · 2025-08-13

We thank the reviewers for their time, effort, and constructive feedback, which strengthened our work through rebuttal and discussion. Below, we provide a summary of the key points from our rebuttal as a final remark.

**Cross-reviewer strengths.** Reviewers highlighted the practical impact for efficient on-device agents [qN16], clear writing and organization [qN16, EVtf, NTqR], and comprehensive evaluation across eight benchmarks [qN16, EVtf, NTqR, yrcR]. Multiple reviewers valued strong out-of-domain generalization [qN16, NTqR, yrcR], the simplicity and effectiveness of FTP and SAG with thorough ablations [NTqR, yrcR], and training small models (0.5B-3B) with limited data [NTqR]. The clarity and reproducibility of our released code [NTqR] were noted; **two reviewers raised scores after discussion [qN16, EVtf]**, others maintained positive assessments.

**Clarifications on our contribution.** [qN16, EVtf] Agent Distillation transfers agentic behavior (reason-act-observe trajectory) of LLM agents into small LMs (0.5B-7B), showing that distilling agentic behavior with retrieval and code tools improves performance on challenging math and factual reasoning tasks across 8 reasoning benchmarks. Two targeted components make it more effective: (1) First-Thought Prefix (FTP) addresses an agent-specific failure mode in reasoning tasks; (2) Self-Consistent Action Generation (SAG) adapts self-consistency to action generation using environment observations, without labels or external judges.

**New results added during rebuttal.**
- SAG-only ablation [qN16]: consistent gains (up to +2.2 avg) across 0.5B-3B.
- Temperature ablation (t=0.2-1.0) [qN16]: flat to mildly positive; 0.4 chosen to avoid determinism.
- AIME robustness [qN16]: 5-seed mean/std show low variance.
- Full FT vs LoRA (1.5B) [qN16]: LoRA outperforms, likely avoiding overfitting.
- Cross-model generalization [NTqR]: We confirm gains of ours over CoT distillation on Llama-3.2-1B and Phi-4-mini.

**Positioning vs prior work.** [EVtf] Our goal differs from function-calling distillation with DPO or tool-learning methods: we target open-ended tools (retrieval, Python) and interactive agentic reasoning across eight challenging benchmarks with smaller models (0.5B-3B). We cited and discussed these works in the revision.

All reviewers acknowledged our contribution. We believe the paper now offers a clear, well-evidenced step toward practical, efficient agents using retrieval and code tools built on small LMs.

---

### Decision · Program_Chairs · 2025-09-17

**Decision:**

Accept (spotlight)

**Comment:**

In this submission the authors introduce Agent Distillation, a new method for distilling model capabilities (such as reasoning and tool-use) from LLM agents into smaller models. Their approach utilizes LLM-generated trajectories of reason-act-observe cycles, and combines 2 approaches: 1) First-Thought Prefix, which enhances the quality of the teacher LLM's trajectories, and 2) Self-Consistent Action Generation, which improves the robustness of the student models at test time. The authors conduct empirical evaluation where they demonstrate the effectiveness of the proposed framework on factual and math reasoning tasks.

The reviewers agree that the idea behind agent distillation is clear and well-motivated, and that the paper itself is well-written and easy to follow. They also agree that the agent distillation offers a practical and effective pathway for transferring the agentic capabilities of LLMs into smaller, more efficient models, showing consistent improvements over baselines. The reviewers also appreciated the inclusion of the code with the submission, as well as cross-domain generalization results.

Overall, the reviewers unanimously agree that the paper provides clear and well-justified methodology, backed by rigorous evaluation. I am therefore pleased to recommend this paper for acceptance.